# Bayesian Optimization over Discrete Structured Inputs by Continuous Objective Relaxation

## Abstract

To optimize efficiently over discrete data from few available target observations is a challenge in Bayesian optimization. We propose a continuous relaxation of the objective function and show that inference and optimization is computationally tractable. The advantages are the continuous treatment of the problem and directly incorporating available prior knowledge over the inputs. Motivated by optimizing expensive biochemical properties from discrete sequences, we consider optimization with few observations and strict budgets. We leverage available and learned distributions from domain models for a weighting of the Hellinger distance, which we show to be a covariance function. Our results include a domain-model likelihood weighted kernel and acquisition function optimization with continuous and discrete algorithms. Lastly, we compare against state-of-the-art Bayesian optimization algorithms on sequence optimization tasks: 25 small-molecule tasks and two protein objectives.

## 1  Introduction

Optimizing discrete inputs with respect to multi-dimensional targets is challenging as gradients are undefined. A common additional difficulty are strict limits on the number of observations and expensive evaluation of the objective, *e.g.* in protein engineering and drug discovery. Bayesian optimization is the *de facto* standard approach for this setting, but most methods in this realm focus on optimizing continuous variables (Močkus, 1975; Shahriari et al., 2016; Garnett, 2022). We focus on optimizing sequences of tokens, *i.e.* a string of amino acids or small molecule descriptors; inspired by the algorithmic demands of protein and drug design (Biswas et al., 2021; Gao et al., 2022). The need to design both proteins and small molecules efficiently has resulted in many different domain models in recent years (Notin et al., 2023; Bagal et al., 2021). In practice, many models are trained in an unsupervised way and can be used as domain-specific latent variable models (Riesselman et al., 2018; Frazer et al., 2021; Detlefsen et al., 2022; Notin et al., 2022). We show how to utilize available probabilistic models over sequences to transform the optimization from discrete sequential inputs to the continuous domain. Specifically, we relax the problem by mapping sequences to distributions and optimize in distribution space, which lets us incorporate prior information directly. Adhering to a strict evaluation budget during optimization and considering realistic sequences requires us to incorporate and leverage strong prior knowledge of the domain.

**In this paper** we first propose a continuous relaxation of the objective function in standard Bayesian optimization (BO), by mapping discrete sequences to the space of probability distributions (Section 3.1). Secondly, we show how inference and optimization remain computationally tractable. This domain transformation allows us to incorporate available prior densities, *e.g.* in the form of an (unsupervised) deep generative models. The result is a covariance function that scales linearly with sequence length (Section 3.2.1). *Empirically*, we demonstrate that our proposed approach performs well when optimizing protein sequences and is on par with established BO methods in optimizing small molecule properties under a very tight evaluation budget. We provide results for 25 small molecule tasks and two protein optimization tasks (Section 4).

## 2  Background

**Problem statement**     Given is a set of discrete sequences of length at most $L \in \mathbb{N}$ and an alphabet of $A$ tokens such that for each position of each sequence, there are $|A|$ possibilities. We define our discrete

input set $\mathbb{X} := \cup_{l=1}^{L} A^l$ as the sequences composed of the tokens up to length $L$. Given a *costly* black-box function $f : \mathbb{X} \mapsto \mathbb{R}$, which provides a value for each input sequence, our objective is to minimize $f$ such that $\boldsymbol{x}_* := \arg\min_{\boldsymbol{x} \in \mathbb{X}} f(\boldsymbol{x})$ – using as few function evaluations as possible. Additionally, there exists a mapping $\phi : \mathbb{X} \mapsto \mathbb{P}$ that can be derived from a set of (unlabeled) input sequences.

One specific example of this are proteins, for which $A$ is the set of naturally occurring 20 amino acids and $\mathbb{X}$ contains all possible protein sequences up to length $L$. The function $f$ is a measurable property of a protein sequence (*e.g.* thermal stability). Acquiring a label is costly because it requires wet-lab experiments to obtain a measurement. Therefore, the initial set of candidates contains only very few labels, which limits our ability to pre-train a supervised surrogate model. This differs from commonly reported BO setups, which may require a large pool of labeled inputs to train a surrogate model prior to the optimization (Gómez-Bombarelli et al., 2018; Tripp et al., 2020; Stanton et al., 2022; Maus et al., 2023; Lee et al., 2024; Ziomek and Bou Ammar, 2023; Kong et al., 2024).

**Bayesian optimization**  consists of a surrogate model $m$ for $f$, and an acquisition function $\alpha$ with the objective to find the global optimum $f^* = \min_{x \in \mathbb{X}} f(x^*)$ (Garnett, 2022), which in our work is set to be the minimum. The model is updated at each iteration given the observations of all experiments and the acquisition function is numerically optimized on the surrogate to select the next evaluation of $f$. The algorithm ends when the function value converges to the optimum or the evaluation budget $b$ is exhausted. Typically, $m$ is a Gaussian process (Rasmussen and Williams, 2006), and popular choices for $\alpha$ are *Expected Improvement* (Jones et al., 1998) and the *Upper Confidence Bound* (Srinivas et al., 2012). BO makes no assumptions about the domain and input space in which we seek the optimum.

**Gaussian process regression**  Gaussian processes (GPs) are a typical choice for $m$ due to their expressiveness and closed-form inference. A GP is a collection of random variables, such that every finite subset follows a multivariate normal distribution (Rasmussen and Williams, 2006, p. 13). The prior is described by a mean function $\mu$ (often set to $\mu(\boldsymbol{x}) := 0$), and a positive definite covariance function (kernel) $k : \mathbb{X} \times \mathbb{X} \mapsto \mathbb{R}$. Assuming that observations of $f$ are distorted by Gaussian noise, the posterior over the function $f$ is again a GP.

For discrete input spaces, continuous numerical optimization algorithms cannot *directly* find the optima of $\alpha$. One approach is to fit a latent variable model and optimize in latent space (Lu et al., 2018; Gómez-Bombarelli et al., 2018), *i.e.* latent BO. Since it is unclear whether the Euclidean distance in representation space is a reliable proxy for similarity (Detlefsen et al., 2022), we develop an approach that is a relaxation through a constrained probability space and uses a distance measure for probability vectors (Section 3.1).

## 2.1 Related Work

We provide an overview of contemporary BO methods that optimize discrete structured inputs over learned latent models, highlighting practical challenges that are worth addressing. Following a discussion of surrogate models on latent representations, strict budgets, and related discrete optimizers, we present the use-case for probabilistic domain models and how it differentiates our contribution.

**GPs over latent representations**  Lu et al. (2018) investigated Bayesian optimization by defining a Gaussian process model directly on the latent space of a variational autoencoder (VAE). Stanton et al. (2022) also formulate a GP surrogate on learned latent variable models. The distance measure in standard covariance functions, *e.g.* the Matérn or Squared-exponential kernel (Rasmussen and Williams, 2006) is not always satisfied in practice, since a learned latent space need not have a Euclidean measure (Arvanitidis et al., 2018). A key assumption for kernels based *just* on Euclidean distance measures is that far-apart observations are independent from one another given a particular length-scale. This assumption is not necessarily fulfilled in learned latent representations, where two sequences can be highly related, while being far away in latent space or close in latent space without being related (Detlefsen et al., 2022).

**Budgets in Bayesian optimization**  Some BO algorithms rely on a significant number of black-box function evaluations either via pre-training surrogates *e.g.* Gómez-Bombarelli et al. use $2\,000$,[1] Tripp et al. uses $10\,000$ observations for GP pre-training, and large labeled pools are presented in Maus et al.; Kong et al.; Lee et al. ($> 50\,000$, $62\,500$, $80\,000$ respectively). This means that these approaches are ruled out by settings

---

[1]Specifically, $250\,000$ (labelled) pre-training samples for the embedding in (Gómez-Bombarelli et al., 2018).

where computational labels are not available and labels are prohibitively expensive, *e.g.* bio-chemical assay experiments (Gao et al., 2022). The problems we consider have few observations available at the start of the optimization and the available budget to evaluate the function is limited *i.e.* $b \ll 10^3$.

**High-dimensional Bayesian optimization** Bayesian optimization in high-dimensional spaces over categorical and mixed-input variables is a large field, recently surveyed by Dreczkowski et al. (2023) and González-Duque et al. (2024). Approaches include graph product kernels, with and without trust regions (Oh et al., 2019; Wan et al., 2021), that leverage embedded linear subspaces, and kernels built for categorical variables (Moss et al., 2020; Papenmeier et al., 2022; 2023), or mapping combinatorial variables to Hamming embeddings (Deshwal et al., 2023). To optimize bio-chemical sequences by learning latent representations has been proposed by Gómez-Bombarelli et al. (2018), and a weighted retraining schemes to represent promising points by Tripp et al. (2020). Maus et al. and Stanton et al. combine surrogate- and representation-learning given a sufficient pre-training pool. Replacing GPs with ensemble methods is done in Gruver et al. for guided diffusion.

**Multi-objective Bayesian optimization** Paria et al. (2020) propose multi-objective Bayesian optimization by scalarizing functions of the objective and a Pareto front objective. These scalarizating functions are commonly linear. This approach is extended in Selega and Campbell (2022) to expectations over the scalarized acquisition and applied to biomedical objectives. (Stanton et al., 2022) also accounts for multiple objectives by optimizing a Pareto front.

Closely related to our work are the articles by Garrido-Merchán and Hernández-Lobato (2020) and Daulton et al. (2022). The former proposes a continuous relaxation for *categorical inputs* on discrete and mixed spaces, whereas we relax the objective. The latter approach introduces a continuous relaxation of the acquisition function in its *probabilistic reparameterization* (PR). PR differs from our approach as it does not consider a constraint probability space of the inputs (which we introduce in Eq. (1) and (2)) and does not include likelihoods of the inputs given a parameterized prior model; the former is required for computational feasibility. The result is that we can optimize problems of significantly higher dimensionality (larger $L$ and $A$ cf. Appendix H).

Our proposed transformation from a discrete optimization problem to a continuous one is at its core a linear programming relaxation (Ge and Huang, 1989; Matoušek and Gärtner, 2007, p. 33). We view this relaxation as optimizing in the space of probability distributions over $\mathbb{X}$. To the best of our knowledge, no current approach formulates BO continuously with a covariance function on distributions over discrete sequence inputs, defining the GP prior over the *objective* that extends to a continuous treatment of the acquisition function. This formulation allows us to map the inherently high-dimensional BO-problem to a constrained probability space (Section 3.1). Furthermore, applying a weighting from distributions to obtain a kernel has not been done prior to this work and incorporates a ranking of the candidates based on prior domain likelihoods (Section 3.2.1).

## 2.2 The case for pre-trained probabilistic domain models

As an alternative to learning a new continuous search space model prior to the optimization, we follow a different approach. We will work directly with probability distributions over the alphabet, which can typically be extracted meaningfully from pre-trained models in the domain. Specifically, advances in bioinformatics and chemoinformatics have resulted in domain models over input sequences, *i.e.* amino-acid (protein) and nucleic-acid (gene) sequences or molecular tokens (Rives et al., 2021; Brixi et al., 2025; Bagal et al., 2021). Apart from large language models (Marin et al., 2023; Notin et al., 2023), one example is a simple protein sequence lookup in a resource like the Protein Data Bank (Berman et al., 2000) done with a hidden Markov model (Eddy, 2011). The scores and likelihoods obtained from these models often correlate with properties of interest (Notin et al., 2023; Lin et al., 2023). Such models can already be used to (i) transform the problem, and (ii) apply a likelihood based ranking over candidates for optimization (see Section 3.4).

# 3 Continuously relaxed Bayesian optimization

As our main contribution we propose a *continuous relaxation of the objective function* which at first glance yields an intractable problem, that requires constraining. We show how to restrict the optimization problem and how to recover computational tractability. This leads us to develop a covariance function that acts on probability distributions and incorporates *a priori* available probability densities for our surrogate function.

### 3.1 From discrete to continuous space

We turn the discrete optimization problem from Section 2 into a continuous one by minimizing the expected function value of $f$ in the space of probability distributions over $\mathbb{X}$. This gives a differentiable function $\bar{f}$ with the same optima:

$$\bar{f}(\mathbf{p}) := \mathbb{E}_{x \sim \mathbf{p}}[f(x)] = \sum_{x \in \mathbb{X}} f(x)\mathbf{p}_x, \tag{1}$$

where $\boldsymbol{p} \in \mathbb{P} := \{\boldsymbol{p} \in [0,1]^{|\mathbb{X}|} \mid \sum_i \boldsymbol{p}_i = 1\}$ are probability distributions over $\mathbb{X}$, *i.e.* real vectors with elements between 0 and 1 whose components sum to 1. Note that each element of $\boldsymbol{x} \in \mathbb{X}$, which is a sequence of tokens, can be represented as $\boldsymbol{p} := \mathbb{1}_{\boldsymbol{x}}$; *i.e.* as Dirac probability vector over the tokens with full mass on $\boldsymbol{x}$, such that $\boldsymbol{p}_{\boldsymbol{x}'} = 1$ iff $\boldsymbol{x} = \boldsymbol{x}'$ and 0 otherwise. The transformation of the problem in Eq. (1) is a linear programming relaxation (Ge and Huang, 1989; Matoušek and Gärtner, 2007, p. 33), which we require for our later contributions.

**Proposition 1.** *Assume that $f$ has a unique optimum in $\boldsymbol{x}_*$, then $\bar{f}$ has a unique optimum in $\mathbb{1}_{\boldsymbol{x}_*}$.*

*Proof.* Deferred to Appendix A. □

We consider the case of multiple optima in Proposition 4 and provide the proof in Appendix A.1.

A continuous objective function with preserved optima may naively appear sufficient for the successful application of BO. However, our relaxation introduces computational challenges that we proceed to resolve.

**Representation** Even for small input sequences (*e.g.* $L < 100$), any $\boldsymbol{p} \in \mathbb{P}$ is infeasibly large. We will have to restrict $\mathbb{P}$ to be able to work with it. Note that the majority of the initial $\mathbb{P}$ space entries are highly unlikely given that inputs are of a particular length and tokens can be positionally conserved – relevant to the objective function. For example, a protein to be optimized has a particular length and is characterized by the occurrence of specific, positionally conserved tokens. We therefore consider $\mathbb{P}_f$, the space of factorizing distributions of length $l$

$$\mathbb{P}_f := \left\{ \mathbf{p} \in [0,1]^{l \times |A|} \middle| \mathbf{p} \geq 0, \forall l : \sum_{a=1}^{|A|} \mathbf{p}_{l,a} = 1 \right\}. \tag{2}$$

This effectively gives us a matrix of likelihoods that at each position sum to one and which we index by position over length and tokens, respectively.

**Inference** A Gaussian process over $f$ naturally induces a model over $\bar{f}$, from the kernel $k'(\boldsymbol{p}, \boldsymbol{q}) := \sum_{\boldsymbol{x}, \boldsymbol{x}' \in \mathbb{X}} \boldsymbol{p}_{\boldsymbol{x}} k(\boldsymbol{x}, \boldsymbol{x}') \boldsymbol{q}_{\boldsymbol{x}'}$. Evaluating this canonical kernel function is intractable as it naively requires $|\mathbb{X}|^2 = (\sum_{l=1}^{L} |A|^l)^2 \approx |A|^{2L}$ operations and thus $\mathcal{O}(|A|^{2L})$ – even if we consider a restriction of $\mathbb{P}$. Section 3.2 resolves this.

**Optimization** Having established a model $m$ over $\bar{f}$ we must determine how to optimize the acquisition function $\alpha_m$. Even though $\alpha_m$ is continuous, the proposed inputs must remain probability distributions, which prevents us from freely using any optimizer;[2] Section 3.3 discusses this.

### 3.2 The model

A GP prior over $f$ induces a Gaussian process belief over $\bar{f}$, yet computing the posterior over $\bar{f}$ is intractable – even when restricted to $\mathbb{P}_f$. The key is to place a GP prior directly over $\bar{f}$ instead of using the induced prior from $f$. This gives us computationally tractable inference.

Generally, it is not the case that the posterior mean of a GP ($\mathbb{E}_{\mathrm{GP}}[\bar{f}(\boldsymbol{p})] := \mathbb{E}[\bar{f}(\boldsymbol{p})|\mathcal{D}]$ for some dataset $\mathcal{D}$) is equal to the weighted sum of posterior means over each atomistic (Dirac) distribution $\mathbb{E}_{\mathrm{GP}}[\bar{f}(\boldsymbol{p})] \neq \sum_{x \in \mathbb{X}} \boldsymbol{p}(\boldsymbol{x}) \mathbb{E}_{\mathrm{GP}}[\bar{f}(\mathbb{1}_x)]$, as potentially expected from *Inference* in Section 3.1. The main challenge is to find a kernel function which can exploit the structural properties of $\mathbb{P}_f$ *e.g.* distances informed by *prior* probability densities, and subsequently discard regions of low-probability.

---

[2]The vector components must be positive and sum to one.

### 3.2.1 The weighted Hellinger kernel

A relevant kernel $k : \mathbb{P}_f \times \mathbb{P}_f \mapsto \mathbb{R}$ can be constructed from the *Hellinger* distance $r$ (Hellinger, 1909)

$$r(\mathbf{p}, \mathbf{q}) := \sqrt{\frac{1}{2} \sum_{x \in \mathbb{X}} \left( \sqrt{\mathbf{p}(x)} - \sqrt{\mathbf{q}(x)} \right)^2}, \tag{3}$$

$$k(\mathbf{p}, \mathbf{q}) := \theta \exp(-\lambda r(\mathbf{p}, \mathbf{q})). \tag{4}$$

We know $r$ to be negative definite (Harandi et al., 2015), and therefore $k$ is a *positive definite* kernel $\forall \theta, \lambda > 0$ (Feragen et al., 2015). Since we restrict $\boldsymbol{p}, \boldsymbol{q}$ to be in $\mathbb{P}_f$, we can evaluate $k(\boldsymbol{p}, \boldsymbol{q})$ in $\mathcal{O}(L|A|)$ time

$$r^2(\mathbf{p}, \mathbf{q}) = 1 - \prod_{l=1}^{L} \sum_{a=1}^{A} \sqrt{\mathbf{p}_{l,a} \mathbf{q}_{l,a}}. \tag{5}$$

This is done by rewriting Eq. (3) – see Appendix Proposition 5.

For any distinct input pair we observe that $\{\boldsymbol{x}, \boldsymbol{x}'\}$: $r(\mathbb{1}_{\boldsymbol{x}}, \mathbb{1}_{\boldsymbol{x}'}) = 1$ for $x \neq x'$, implying that a Hellinger distance kernel is not necessarily a useful guide for optimization. However, we recall from Section 2.2 that there often exists a *prior ranking* over elements of $\mathbb{X}$ in form of a probability distribution, such as hidden Markov models, variational autoencoders, or large language models with which we can compute likelihoods (Durbin et al., 1998; Riesselman et al., 2018; Frazer et al., 2021; Rives et al., 2021). To use this prior knowledge, we propose to weigh the Hellinger distance using a given ranking. For every positive weighting $w : \mathbb{X} \mapsto \mathbb{R}_+$, we define the weighted distance as

$$r_w^2(\mathbf{p}, \mathbf{q}) := \frac{1}{2} \sum_{\boldsymbol{x} \in \mathbb{X}} w(\boldsymbol{x}) \left( \sqrt{\mathbf{p}(\boldsymbol{x})} - \sqrt{\mathbf{q}(\boldsymbol{x})} \right)^2. \tag{6}$$

**Proposition 2.** *The squared weighted Hellinger distance is negative definite.*

*Proof.* To show that Eq. (6) gives rise to a kernel function we extend the proof by Harandi et al. (2015) which can be found in Appendix B. $\square$

**Proposition 3.** *For product measures $\boldsymbol{p}, \boldsymbol{q}, \boldsymbol{w} \in \mathbb{P}_f$, the weighted Hellinger distance remains linear time computable*

$$r^2(\boldsymbol{p}, \boldsymbol{q}, \boldsymbol{w}) = \prod_{l=1}^{L} \frac{1}{2} \sum_{a_l=1}^{A} \left( \boldsymbol{p}_{a_l,l} \boldsymbol{w}_{a_l,l} + \boldsymbol{q}_{\boldsymbol{a}_l,l} \boldsymbol{w}_{\boldsymbol{a}_l,l} \right) - \prod_{l=1}^{L} \sum_{a_l=1}^{A} \boldsymbol{w}_{a_l,l} \sqrt{\boldsymbol{p}_{a_l,l} \boldsymbol{q}_{a_l,l}}. \tag{7}$$

*Proof.* Deferred to Appendix D.1. $\square$

With the weighted distance, distinct sequences $\boldsymbol{x}, \boldsymbol{x}'$ evaluate to $r_w(\mathbb{1}_{\boldsymbol{x}}, \mathbb{1}_{\boldsymbol{x}'}) = \frac{1}{2}(w(\boldsymbol{x}) + w(\boldsymbol{x}'))$. Sequences with low weighting are considered similar, whereas sequences with high weighting are more independent. Because our setting is data-limited, we require strong modeling assumptions to make optimization feasible; the independence assumption for dissimilar, high-weight sequences is one such choice.

The kernel is particularly suitable to model functions that have a particular threshold or where sets of inputs yield a particular value, *i.e.* in some optimization campaigns we care about an improvement given a reference. If the sequence weighting is 0 (or close to 0), thus highly unlikely according to the weighting model—assuming reasonable correlation between the weighting and the objective—we expect low interest function values. Other zero weighted sequences correlate perfectly under this kernel, allowing us to disregard this vast space with one function evaluation in our BO routine. In practice, we rely on the assumption that domain-model likelihoods correlate with objective values, an assumption supported by established benchmarks such as ProteinGym (Notin et al., 2024), where likelihoods have been shown to provide meaningful rankings of candidate sequences.

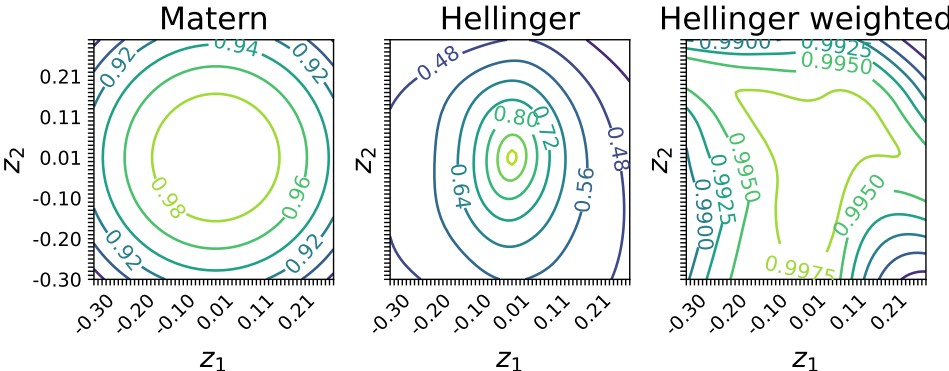

Figure 1: Visual comparison of the (weighted) Hellinger distance kernel and a Matérn 5/2. We use an adaptation of the decoder proposed by Brookes et al. (2019) (see Fig. A1). For the Matérn kernel we visualize $k(0, [z_1, z_2])$ whereas for the Hellinger kernel $k(P(x \mid 0), P(x' \mid [z_1, z_2]))$. With the Hellinger kernel, the decoder induces a more complex, non-Euclidean similarity measure on the latent space, which is non-stationary in the latent space — see Appendix Fig. A2.

**Valid distributions** $w$    Any distribution over a sequence of available input tokens can be used as a weighting distribution, if it can be factorized over the length of the inputs. Since the acquisition depends on this weighting, the likelihood of the inputs should correlate with the downstream objective function. We consider HMMs (Durbin et al., 1998) and VAEs (Kingma and Welling, 2019)—assuming factorizing distributions. Importantly, the weighting distribution can be distinct from the distribution objects in $\mathbb{P}_f$. While in this work the weighting and the model that parameterizes the representation are the same, this is not a requirement. When choosing $w$ it is important that the weighting allows for efficient evaluations of $r_w$. Fig. 1 visualizes how the combination of Hellinger kernel and decoder induces a more complex, non-Euclidean similarity measure on the latent space. The type of model determines the optimization strategy (Section 3.3).

**Kernel specifications**    The weighted Hellinger kernel gives rise to expressive GP models by the product property of kernels (Rasmussen and Williams, 2006, p. 95). Specifically, for a given latent variable model, we propose to use $k'(x, x') := \prod_{n=1}^{N} k_{P(X|Z_n)}(x, x')$, with $N$ as the size of the unlabelled pre-training dataset and $Z_n$ as subsets of the latent space[3] The same hyper-parameters $\theta$ and $\lambda$ apply for each kernel. This yields a product kernel over the available samples. We defer specific choices for $Z_n$ to Section 4. We follow Jones et al. (1998) for hyper-parameter settings and set a constant prior mean function with $\mu := \frac{\mathbf{1}^\intercal K^{-1} y}{y^\intercal K^{-1} y}$ and the amplitude of the kernel to $\theta := \frac{1}{N-1}(y - \mu)^\intercal K^{-1}(y - \mu)$. We set $\lambda$ and $\sigma^2$ by maximizing the evidence $\log p(y|\lambda, \sigma^2)$. Given $N$ pre-training samples and $n = |\mathcal{D}_t|$ as the size of the dataset at a given timestep $t$ of the BO algorithm, we obtain $\mathcal{O}(n^2 N L |A|)$ for the cost of the complete Gram matrix construction and the usual $\mathcal{O}(n^3)$ for the Cholesky decomposition when computing the posterior predictive. Lastly, the choice of a product kernel aggregate emphasizes the dependence on the weighting function. Other valid modeling choices exist that rely on different weighting contributions Jebara et al. (2004).

### 3.3    Optimizing the acquisition function

As a consequence of the continuous relaxation, the acquisition function also acts on the probability distributions. We can find local optima by

   i) discrete optimization with the acquisition function evaluating sequences individually as indicator functions,

   ii) a continuous parameterization of $\mathbb{P}_f$, a) given a ($D$-dimensional) probabilistic decoder mapping latent vectors $\boldsymbol{z} \in \mathbb{Z}^D$ to the space of factorizing distributions   $\mathrm{Dec} : \mathbb{R}^D \mapsto \mathbb{P}_f$, and b) identifying each $P \in \mathbb{P}_f$ through the canonical softmax mapping $\mathrm{Dec} : \mathbb{R}^{L \times A} \mapsto \mathbb{P}_f$ with $\boldsymbol{z}_{l,a} \mapsto \frac{\exp(\boldsymbol{z}_{l,a})}{\sum_{a'=1}^{A} \exp(\boldsymbol{z}_{l,a'})}$ for $l \in [1, \dots, L]$ and $a \in [1, \dots, A]$, which enables a continuous optimization of $\beta(\boldsymbol{z}) := \alpha(\mathrm{Dec}[\boldsymbol{z}])$,

---

[3]Specifically, given some latent encoder $\mathrm{enc} : \mathbb{X} \mapsto \mathbb{R}^D$ we obtain latent vectors $Z_n = \mathrm{enc}(\boldsymbol{x})$ with which to construct $P(X|Z_n)$.

iii) manifold optimization on $\mathbb{P}_f$, since the space $\mathbb{P}_f$ is a product simplex, and we can thus use any manifold optimization algorithm directly on the acquisition function (Boumal, 2014).

Combinations of these options apply. For the continuous case we ideally want the optimization to terminate with an atomic distribution such that the decision of which sequence to evaluate is unambiguous. Even if this is not the case, the optimization narrows down the choice of candidates and we can choose the most likely sequence or sample from the optimized distribution and use the acquisition function to score the sampled candidates. In this work we focus on the *discrete* and *continuous* case and leave the exploration of manifold optimization for future work.

### 3.4 The `CoRel` algorithms

Algorithm 1 shows the continuous relaxation (`CoRel`) in the general Bayesian optimization loop given a continuous parameterization of $\mathbb{P}_f$, here the decoder of a VAE. See Algorithms 3 and 4 in the Appendix for discrete or direct continuous optimization.

Given is a set with $n_0$ starting sequences with observations $\mathcal{D}_0$, an acquisition function $\alpha$ on $\mathbb{P}_f$, and a model that defines the weighting. We fit a GP posterior predictive by optimizing the kernel parameters (see Alg. 1). The acquisition function is queried to find the maximizing probability vector with the predictive GP on the parameterized space of $D$ dimensions, given some probabilistic parameterization $\phi$. From the probability vector we can obtain a sequence or a set of sequences within a given budget (see Alg. 2). The larger the internal sampling budget ($b$) the more sequences proposals will be evaluated. Ultimately, the black-box function is evaluated on the sequence and the observations and inputs are added to the dataset (see Alg. 1). This is repeated until the black-box budget is exhausted.

---

**Algorithm 1** `CoRel` with parameterized optimization

**Input:** acquisition $\alpha : \mathbb{P}_f \mapsto \mathbb{R}$, black-box $f : \mathbb{X} \mapsto \mathbb{R}$, $\mathcal{D}_0 = \{\boldsymbol{x}_i, \boldsymbol{y}_i\}_{i=1}^{n_0}$, LVM $\phi : \mathbb{R}^D \mapsto \mathbb{P}_f$, budgets $t_{\max}, b$
**Output:** $\mathcal{D}_{t_{\max}}$
    **for** $t \in 1, ..., t_{\max}$ **do**
        $m \leftarrow \text{trainModel}((\mathbb{1}_{\boldsymbol{x}_i}, \boldsymbol{y}_i)_{i=1}^{t})$
        $\boldsymbol{z}_* \leftarrow \arg\max_{\boldsymbol{z}} \alpha(\phi(\boldsymbol{z}), m)$
        $\boldsymbol{p}_* \leftarrow \phi(\boldsymbol{z}_*)$
        $\boldsymbol{x} \leftarrow \text{getSequenceFromDistribution}(\boldsymbol{p}_*, \alpha, b)$
        $\mathcal{D}_{t+1} \leftarrow \mathcal{D}_t \cup \{\boldsymbol{x}, f(\boldsymbol{x})\}$
    **end for**

**Algorithm 2** getSequenceFromDistribution

**Input:** distribution $\mathbb{P}$, acquisition $\alpha$, evaluations $b$
**Output:** $\boldsymbol{x}_*$
    $\boldsymbol{x}_* \leftarrow \arg\max_{\boldsymbol{x}} \mathbb{P}(\boldsymbol{x})$
    $y_* \leftarrow \alpha(\mathbb{1}_{\boldsymbol{x}_*})$
    **for** $t \in 1, ..., b$ **do**
        $\boldsymbol{x} \leftarrow \boldsymbol{x} \sim \mathbb{P}$
        $y \leftarrow \alpha(\mathbb{1}_{\boldsymbol{x}})$
        **if** $y > y_*$ **then**
            $y_* \leftarrow y, \boldsymbol{x}_* \leftarrow \boldsymbol{x}$
        **end if**
    **end for**

---

The key contributions in the two algorithms are (i) the use of factorizing distributions through a prior model $\phi$, (ii) the surrogate model with a prior weighting, and (iii) the acquisition function acting on $\mathbb{P}_f$. While the use of an $\arg\max$ can be considered a common choice to recover a sequence, our approach allows us to sample sequences from the distribution object.

### 3.5 Convergence

A discrete optimizer together with UCB as the acquisition function can achieve sublinear regret under the conditions laid out in Srinivas et al. (2012). We note, however, that the regret analysis of Srinivas et al. does not directly extend to our weighted Hellinger kernel, which is data-dependent and would require a new analysis of its RKHS and information-gain properties. Developing such bounds is beyond the scope of this work, and in practice our algorithm employs Expected Improvement rather than UCB, for which only limited asymptotic convergence results exist; we therefore focus on empirical evaluation. The continuous optimization case is a distinct problem outside the scope of this paper and additional considerations are in Appendix G.

## 4 Empirical results

We benchmark our method to optimize i) several Red Fluorescent Proteins (RFP) as proposed by Stanton et al. (`LamBO`), ii) an enhanced Green Fluorescent Protein (eGFP) with a VAE proposed by Brookes et al. (`CBas`) as a fitness proxy, and iii) 25 molecular optimization tasks (PMO) proposed by Gao et al. (see Appendix H). The first task (RFP) requires us to optimize both the stability and surface accessibility of protein candidates by modifying the tokenized sequence of amino acids and evaluates established domain oracles (see Appendix I.3). The second (GFP) task evaluates the fitness surrogates in Brookes et al. (2019) as a proxy for green protein fluorescence. The last set of multiple tasks (PMO) are a benchmark for label-efficient small molecule optimization based on property optimization, (re-)discovery of molecules, docking proxies, and other tasks. We solve problems i)-iii) with `CoRel` with $\mathcal{D}_0$ denoting the initial set of labeled candidates that define a Pareto front, *i.e.* six RFP sequences, three GFP sequences, and one small molecule for each PMO task. This setup is motivated by drug-discovery tasks where initial experimental observations can be prohibitively expensive. We keep pre-training data equal between all tested methods and fix the prior domain model $\phi$ for a fair comparison.[4]

### 4.1 Continuous optimization

**Optimizing with a latent variable model** lends itself to `CoRel` as a continuous optimizer. Brookes et al. (2019) provide a continuous parameterization of $\mathbb{P}_f$ to solve the GFP problem in the form of a pre-trained latent decoder (details are in Appendix I.4). The `CBas` oracle function evaluations serve as surrogate for the true GFP fluorescence values. To qualitatively inspect our method we evaluate the covariance function values of the (2D) latent space in an area around the reference sequence. Fig. 1 shows the evaluated Hellinger (Eq. (3)) and weighted Hellinger kernel (Eq. (6)), which are not equidistant in latent space, like the Matérn kernel evaluations, and show the contribution of the probabilistic model weighting distribution to the covariance function values. These values expectedly change when computed with respect to a different reference point and higher covariance values are assigned in a density around the reference points and the respective decoding probabilities (Appendix Fig. A2).

We optimize GFP sequences with `CoRel`, defining the product kernel model by setting $Z_n$ as the VAEs latent training samples. We compare against greedy random selection of mutations close to the reference sequences (`random HC`), Probabilistic Reparameterization (`PR`) (Daulton et al., 2022) and random sampling of the sequence (Sobol). We run `CoRel` with (i) a product kernel from all available unlabelled data and (ii) a $N = 5\,000$ uniformly sampled subset, both use a continuous optimization and Expected Improvement acquisition. Fig. 4 shows that we find larger objective values within the allotted budget (100 queries) compared to `random HC`. The full product kernel (i) is terminated after 60h compute time,[5] and the sampled subset-kernel runs for less than 30h. Sobol and `PR` are overlying and neither propose an improvement over the starting sequences. `CoRel` prioritizes extreme values of the oracle (Appendix Fig. A6).

To optimize small molecule (SELFIE) tokens we solve 25 PMO tasks continuously with `CoRel`, defining a product kernel from a sampled set of (uniform sampled $N = 1\,000$ unlabelled) training sequences, testing against `PR` (Daulton et al., 2022), `Bounce` (Papenmeier et al., 2023), `Turbo` (Eriksson et al., 2019), `VanillaBO` (Hvarfner et al., 2024), `CMA-ES` – see Table 1 for a summary grouped by tasks (an unaggregated overview is in Table A3). The VAE ($\phi$) which is used for all methods is trained on ZINC250k SELFIES (details in Appendix I.6). Since single property optimization tasks tend to dominate result aggregation (`logP`, `QED`, `SA`) we report grouped tasks, as to not overrepresent them. Again we optimize continuously with an EI acquisition function. `CoRel` obtains the best performance for `qed` optimization and the `gsk3` docking task compared to other methods. However, `CoRel` is not consistently competitive across all tasks, suggesting that different, potentially task-dependent, priors should be investigated for small molecules. We find, surprisingly, that `CMA-ES` dominates the majority of tasks, which we discuss in Appendix I.11.

---

[4]Specifically, a (RFP) HMM computed from 250 sequences, VAE from 250 000 unlabelled samples (ZINC250k, PMO).
[5]At this point 26 iterations are completed for all seeds.

Table 1: PMO 25 tasks aggregated by group, mean ±standard error (SE) across tasks is reported; each value is the mean over the best observations (9 seeds), normalized by the best value per task. For a complete overview see Appendix Table A3.

| group | BO | | | | | | | | | | references | | | |
| | CoRel | | PR | | Bounce | | Turbo | | VanillaBO | | CMA-ES | | random HC | |
| | value | ±SE | value | ±SE | value | ±SE | value | ±SE | value | ±SE | value | ±SE | value | ±SE |
|---|---|---|---|---|---|---|---|---|---|---|---|---|---|---|
| optimize | 0.90 | 0.03 | 0.78 | 0.05 | 0.72 | 0.16 | **0.94** | 0.08 | 0.84 | 0.05 | 0.93 | 0.05 | 0.83 | 0.01 |
| discover | 0.88 | 0.02 | 0.61 | 0.10 | 0.09 | 0.03 | 0.98 | 0.12 | 0.91 | 0.06 | **0.99** | 0.09 | 0.85 | 0.09 |
| dock | **0.60** | 0.13 | 0.22 | 0.04 | 0.21 | 0.21 | 0.57 | 0.18 | 0.45 | 0.20 | 0.60 | 0.24 | 0.46 | 0.26 |
| mpo | 0.67 | 0.08 | 0.37 | 0.14 | 0.19 | 0.06 | 0.92 | 0.10 | 0.83 | 0.08 | **0.98** | 0.11 | 0.79 | 0.16 |
| other | 0.63 | 0.05 | 0.54 | / | 0.28 | 0.07 | 0.83 | 0.08 | 0.80 | 0.08 | **0.86** | 0.06 | 0.79 | 0.09 |

## 4.2 Discrete optimization

**Optimizing in the discrete proposed sequence setting**   generates proposals directly in the sequence space. We optimize the RFP problem with respect to two properties: stability and surface area accessibility (SASA) proposed by Stanton et al. (2022). We use `LamBO` as a state of the art reference in this optimization setting and also compare against a greedy hill-climb random sequence mutations.[6] We take the Pareto front (six reference RFP sequences) as $\mathcal{D}_0$ which differs from the larger pre-training set in (Stanton et al., 2022) and set the oracle evaluations $t_{\max}$ to 180 queries. To optimize for multiple tasks in the BO algorithm we use the expected hypervolume improvement (EHVI) as acquisition function (Daulton et al., 2020). To build the $\mathbb{P}_f$-space we take a hidden Markov model (HMM), obtained from HMMER (Eddy, 2011). The HMM serves as our $\phi$ model to parameterize our distributions. The choice for the HMM as $\phi$ is motivated by the process of querying for related sequences, *e.g.* the set of starting candidates, which already yields such a model. This step is required when setting up the initial RFP problem-set (Stanton et al., 2022).[7] Akin to the optimization done in `LamBO` we mutate the elements of the input sequences to maximize EHVI acquisition values. Fig. 2 shows that running `CoRel` obtains a larger relative hypervolume compared to `LamBO`, and `random HC` which modifies two residues selected at random from elements in the Pareto front and retains the best results for subsequent iterations. This results in a larger Pareto front of the respective protein candidates (Fig. 3). In the setup by Stanton et al. (2022) where over 500 initial sequences are available we achieve on-par performance (Fig. A4). However, we find that if significantly more starting candidates are available, a larger Pareto front is optimized and `LamBO` outperforms our method (see Fig. A3).

## 5   Discussion

**A constrained high-dimensional problem is a solvable problem**   The experiments we consider are very high-dimensional and the proposed `CoRel` approach constrains this problem to make it computable. Other relaxations that are unconstrained can become unsolvable (Daulton et al., 2022). Although constraining other methods is an option, there exists no generally applicable recipe for other high dimensional solvers. We obtain this computability, in part, due to the factorizing assumption underlying $\mathbb{P}_f$ (Eq. (2)). While this modeling assumption holds for inputs with independent tokens, it fails for co-dependencies in the inputs, *e.g.* long-range residue dependence or small molecule substructures and is thus limited. Furthermore, not all discrete problems can assume inputs of consistent length and positional, problem-specific conservation of tokens – that is, consistent structure over inputs. Both the length consistency and a prior model over token occurrence are requirements for our approach.

**We focus on the surrogate model rather than the acquisition function**   Advances in Bayesian optimization can be achieved by building a useful surrogate function to model $f$ or by investigating the acquisition function. In addition to the problem transformation our primary contribution is the surrogate model through the covariance function. `CoRel` focuses on the properties of the GP, which incorporates a ranking over the continuously relaxed inputs. The surrogate model we obtain predicts points of interest to

---

[6]`random HC` mutates a few positions at random and retains best mutations per iteration.
[7]The RFP data includes a wide range of additional sequences that are not in the initial Pareto front (see Appendix I.5).

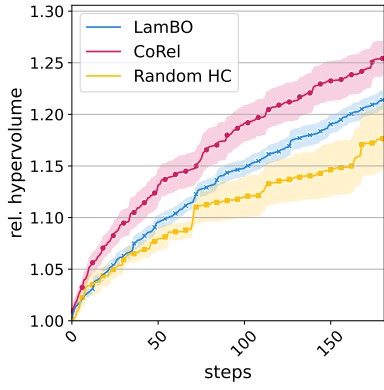 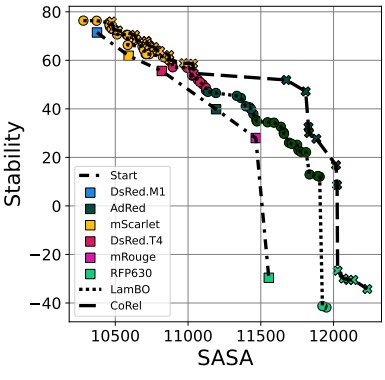 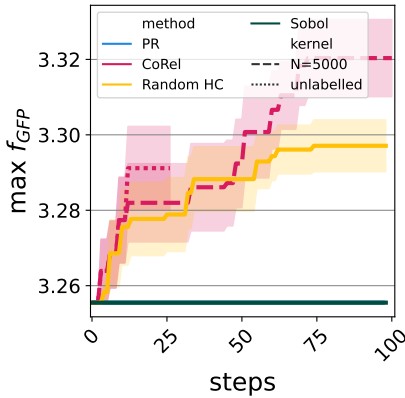

Figure 2: The RFP Pareto front is optimized and relative hypervolume computed respective the six labelled starting sequences. Reported is mean ±SE (shaded) across 9 seeds (random 5).

Figure 3: Pareto front of the RFP stability and surface accessibility (SASA). Initial sequences (■), the best proposals by LamBO (●) and CoRel (✖) (9 seeds).

Figure 4: GFP optimization on CBas. `CoRel` has $k$ all sequences, and $N = 5\,000$ subset. Reported is mean (line) ±SE (shaded) (`CoRel` 8 seeds). `PR` and Sobol (9 seeds) remain at ≈ 3.25.

observe, at the cost of capturing the underlying function landscape. An investigation of acquisition functions are out of the scope for this work and we rely on the established results in the field (Jones et al., 1998; Daulton et al., 2020).

**Other distance measures apply**  Any distance metric on probability vectors applies as long as the resulting kernel is valid. To be able to utilize a kernel we require it to be (i) positive semi-definite, (ii) efficient to compute, such that it scales to many samples, and (iii) it can be weighted with likelihoods. The Hellinger kernel satisfies these criteria. It is directly defined for the space of factorizing distributions $\mathbb{P}_f$ and therefore directly leverages the probability vectors as described in Section 3.2. While other metrics for probability vectors exist (*e.g.* the Jensen Shannon Divergence or Wasserstein-1 distance (Menéndez et al., 1997)), their induced kernels are usually not as efficient to compute or to weight - making them impractical for our setting. `CoRel` can potentially work with either; however we emphasize the linear runtime of our kernel, which may not translate to alternatives.

**Relying on prior models**  Given that distributions over discrete input elements exist that have been shown to work for a particular problem (Notin et al., 2023), they can be used directly with `CoRel`. The assumption that a well-defined model exists for the problem domain is crucial and our ability to find good solutions depends on the choices for relaxation parameterization and weighting function. That means if no prior model over the input tokens exists with which to derive a probability distribution, the proposed problem transformation is not solvable. However, the lack of a model implies that no prior knowledge exists and the problem has to be treated naively. Furthermore, if the weighting function is only weakly correlated – or entirely uncorrelated – with the target function, then the kernel cannot guide candidate selection effectively. In this case, the procedure reduces to essentially random sampling of the objective.

## 6 Conclusion

We have shown an approach to cast discrete Bayesian optimization problems as continuous with a computationally tractable, nonpathological choice of kernel function. Our approach allows us to leverage domain knowledge from prior unsupervised models for Bayesian optimization, and the empirical assessment has demonstrated the applicability to biochemical problems across several diverse tasks. We have transformed an initially infeasible problem space and demonstrated performance on particularly challenging formulations of optimization problems with very few starting observations and strict budgets.

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

# A  Proof that the relaxed objective has the same optima

**Proposition 1.** *Assume that $f$ has a unique optimum in $\boldsymbol{x}_*$, then $\bar{f}$ has a unique optimum in $\mathbb{1}_{\boldsymbol{x}_*}$.*

*Proof.*

$$\bar{f}(\boldsymbol{p}) = \sum_{\boldsymbol{x} \in \mathbb{X}} f(\boldsymbol{x}) p(\boldsymbol{x}) \leq \sum_{\boldsymbol{x} \in \mathbb{X}} \max_{\boldsymbol{x}'} f(\boldsymbol{x}') p(\boldsymbol{x}) = \max_{\boldsymbol{x}'} f(\boldsymbol{x}') \sum_{\boldsymbol{x} \in \mathbb{X}} p(\boldsymbol{x}) = \max_{\boldsymbol{x}'} f(\boldsymbol{x}') \tag{8}$$

On the other hand: for an optimal $\boldsymbol{x}_*$, *i.e.* $f(\boldsymbol{x}_*) = \max_{\boldsymbol{x}'} f(\boldsymbol{x}')$ *and choose* $\boldsymbol{p}_*(\boldsymbol{x}) := \mathbb{1}_{[\boldsymbol{x}=\boldsymbol{x}_*]}$, *then* $\bar{f}(\boldsymbol{p}_*) = \sum_{\boldsymbol{x} \in \mathbb{X}} f(\boldsymbol{x}) p_*(\boldsymbol{x}) f(\boldsymbol{x}) = \max_{\boldsymbol{x}'} f(\boldsymbol{x}')$. *Since by assumption* $\boldsymbol{x}_*$ *is a unique optimum of* $f$, *this completes the proof.* $\qquad\square$

## A.1  Multiple Optima

**Proposition 4.** *For every $\boldsymbol{p}_*$ with $\bar{f}(\boldsymbol{p}_*) = \max_{\boldsymbol{p}} \bar{f}(\boldsymbol{p})$, if $\boldsymbol{x} \sim \boldsymbol{p}_*$, then $f(\boldsymbol{x}) = \max_{\boldsymbol{x}'} f(\boldsymbol{x}')$.*

*Proof.* Let $\boldsymbol{p}_*$ be s.t. $\bar{f}(\boldsymbol{p}_*) = \max_{\boldsymbol{p}} f(\boldsymbol{p})$ and let $\boldsymbol{x}$ be a sample from $\boldsymbol{p}_*$. From the previous proof we know that $\max_{\boldsymbol{p}} \bar{f}(\boldsymbol{p}) = \max_{\boldsymbol{x}'} f(\boldsymbol{x}')$. Furthermore, since $\boldsymbol{x}$ is a sample from $\boldsymbol{p}_*$, we have that $\boldsymbol{p}_*(\boldsymbol{x}) > 0$. We will prove the proposition by contradiction. Assume $f(\boldsymbol{x}) < \max_{\boldsymbol{x}'} f(\boldsymbol{x}')$, then

$$\max_{\boldsymbol{x}'} f(\boldsymbol{x}') = \bar{f}(\boldsymbol{p}_*) \qquad\qquad \text{previous proof in Appendix A} \tag{9}$$

$$= \boldsymbol{p}_*(\boldsymbol{x}) f(\boldsymbol{x}) + \sum_{\boldsymbol{x}' \neq \boldsymbol{x}} \boldsymbol{p}_*(\boldsymbol{x}') f(\boldsymbol{x}') \qquad \text{definition of } \bar{f} \text{ and separating terms} \tag{10}$$

$$\leq \boldsymbol{p}_*(\boldsymbol{x}) f(\boldsymbol{x}) + \sum_{\boldsymbol{x}' \neq \boldsymbol{x}} \boldsymbol{p}_*(\boldsymbol{x}')) \max_{\boldsymbol{x}''} f(\boldsymbol{x}'') \qquad \text{over estimating all } f(\boldsymbol{x}') \tag{11}$$

$$= \boldsymbol{p}_*(\boldsymbol{x}) f(\boldsymbol{x}) + (1 - \boldsymbol{p}_*(\boldsymbol{x})) \max_{\boldsymbol{x}'} f(\boldsymbol{x}') \qquad \text{simplifying the sum} \tag{12}$$

$$< \boldsymbol{p}_*(\boldsymbol{x}) \max_{\boldsymbol{x}'} f(\boldsymbol{x}') + (1 - \boldsymbol{p}_*(\boldsymbol{x})) \max_{\boldsymbol{x}'} f(\boldsymbol{x}') \qquad \text{assumption on } f(\boldsymbol{x}) \text{ and } \boldsymbol{p}(\boldsymbol{x}_*) > 0 \tag{13}$$

$$= \max_{\boldsymbol{x}'} f(\boldsymbol{x}') \tag{14}$$

This is a contradiction, so we must have $f(\boldsymbol{x}) = \max_{\boldsymbol{x}'} f(\boldsymbol{x}')$. $\qquad\square$

# B  Proof that the weighted Hellinger distance is negative definite

Proof of Proposition 2.

*Proof.* The proof follows Harandi et al. (2015). By definition of *negative definite* we have to show $\forall N \in \mathbb{N}, \forall c_1, \ldots, c_N \in \mathbb{R} : \sum_{n=1}^{N} c_n = 0 \Rightarrow \sum_{n,m=1}^{N} c_n c_m \mathrm{HD}_{\boldsymbol{p}}(\boldsymbol{q}_n, \boldsymbol{q}_m)^2 \leq 0$ .

Let $N \in \mathbb{N}$ and $c_1, \ldots, c_N \in \mathbb{R}$ s.t. $\sum_{n=1}^{N} c_n = 0$.

$$\sum_{n,m=1}^{N} c_n c_m \mathrm{HD}_{\boldsymbol{p}}(\boldsymbol{q}_n, \boldsymbol{q}_m)^2 \tag{15}$$

$$= \frac{1}{2} \sum_{n,m=1}^{N} c_n c_m \sum_{\boldsymbol{x} \in \mathbb{X}} \boldsymbol{p}(\boldsymbol{x}) \left( \sqrt{\boldsymbol{q}_n(\boldsymbol{x})} - \sqrt{\boldsymbol{q}_m(\boldsymbol{x})} \right)^2 \tag{16}$$

∥ *by definition*

$$= \frac{1}{2} \sum_{n,m=1}^{N} c_n c_m \sum_{\boldsymbol{x} \in \mathbb{X}} \boldsymbol{p}(\boldsymbol{x}) \left( \boldsymbol{q}_n(\boldsymbol{x}) + \boldsymbol{q}_m(\boldsymbol{x}) - 2\sqrt{\boldsymbol{q}_n(\boldsymbol{x})\boldsymbol{q}_m(\boldsymbol{x})} \right) \tag{17}$$

∥ *expanding the square*

$$= \sum_{\boldsymbol{x} \in \mathbb{X}} \boldsymbol{p}(\boldsymbol{x}) \left( \sum_{n=1}^{N} c_n \boldsymbol{q}_n(\boldsymbol{x}) \sum_{m=1}^{N} c_m + \sum_{m=1}^{N} c_m \boldsymbol{q}_m(\boldsymbol{x}) \sum_{n=1}^{N} c_n - \sum_{n=1}^{N} c_n \sqrt{\boldsymbol{q}_n(\boldsymbol{x})} \sum_{m=1}^{N} c_m \sqrt{\boldsymbol{q}_m(\boldsymbol{x})} \right) \tag{18}$$

∥ *changing order of summation*

$$= -\sum_{\boldsymbol{x} \in \mathbb{X}} \boldsymbol{p}(\boldsymbol{x}) \sum_{n=1}^{N} c_n \sqrt{\boldsymbol{q}_n(\boldsymbol{x})} \sum_{m=1}^{N} c_m \sqrt{\boldsymbol{q}_m(\boldsymbol{x})} \tag{19}$$

∥ *since* $\sum_{n=1}^{N} c_n = 0$

$$= -\sum_{\boldsymbol{x} \in \mathbb{X}} \boldsymbol{p}(\boldsymbol{x}) \left( \sum_{n=1}^{N} c_n \sqrt{\boldsymbol{q}_n(\boldsymbol{x})} \right)^2 \tag{20}$$

∥ *writing the identical sums over n and m as a square*

$$\leq 0 \tag{21}$$

□

To show that the square-root of the tilted Hellinger distance is a kernel, we follow the same reasoning as in Harandi et al. (2015).

## C  Proof for the efficient evaluation of the Hellinger distance

**Proposition 5.** *For product measures* $\boldsymbol{p}, \boldsymbol{q} \in \mathbb{P}_f$, *the Hellinger distance can be written as*

$$HD(\boldsymbol{p}, \boldsymbol{q})^2 = 1 - \prod_{l=1}^{L} \sum_{a_l=1}^{A} \sqrt{\boldsymbol{p}[a_l, l]\boldsymbol{q}[a_l, l]}$$

.

*Proof.*

$$\text{HD}(\boldsymbol{p}, \boldsymbol{q})^2 = \frac{1}{2} \sum_{\boldsymbol{x} \in \mathbb{X}} \left( \sqrt{\boldsymbol{p}(\boldsymbol{x})} - \sqrt{\boldsymbol{q}(\boldsymbol{x})} \right)^2 \tag{22}$$

*// expanding the square and using that $\boldsymbol{p}$ and $\boldsymbol{q}$ sum to 1.*

$$= 1 - \sum_{\boldsymbol{x} \in \mathbb{X}} \sqrt{\boldsymbol{p}(\boldsymbol{x})\boldsymbol{q}(\boldsymbol{x})} \tag{23}$$

*// property of the Hellinger distance*

$$= 1 - \underbrace{\sum_{a=1}^{A} \cdots \sum_{a=1}^{A}}_{L \text{ times}} \sqrt{\boldsymbol{p}(a_1, \ldots, a_L)\boldsymbol{q}(a_1, \ldots, a_L)} \tag{24}$$

*// rewriting the sum*

$$= 1 - \underbrace{\sum_{a_1=1}^{A} \cdots \sum_{a_L=1}^{A}}_{L \text{ times}} \sqrt{\prod_{l=1}^{L} \boldsymbol{p}[a_l, l]\boldsymbol{q}[a_l, l]} \tag{25}$$

*// using $\boldsymbol{p}, \boldsymbol{q} \in \mathbb{P}_f$*

$$= 1 - \underbrace{\sum_{a_1=1}^{A} \cdots \sum_{a_L=1}^{A}}_{L \text{ times}} \prod_{l=1}^{L} \sqrt{\boldsymbol{p}[a_l, l]\boldsymbol{q}[a_l, l]} \tag{26}$$

*// moving the square-root*

$$= 1 - \sum_{a_1=1}^{A} \sqrt{\boldsymbol{p}[a_1, 1]\boldsymbol{q}[a_1, 1]} \cdot \ldots \cdot \sum_{a_L=1}^{A} \sqrt{\boldsymbol{p}[a_L, L]\boldsymbol{q}[a_L, L]} \tag{27}$$

*// moving unaffected parts of the product out of the sum*

$$= 1 - \prod_{l=1}^{L} \sum_{a_l=1}^{A} \sqrt{\boldsymbol{p}[a_l, l]\boldsymbol{q}[a_l, l]} \tag{28}$$

*// rearranging*

$\square$

## D  Efficient evaluation of the weighted Hellinger distance

### D.1  Product measures

For product measures $\boldsymbol{p}, \boldsymbol{q}, \boldsymbol{r} \in \mathbb{P}_f$, the weighted Hellinger distance can be written as

$$\text{HD}(\boldsymbol{p}, \boldsymbol{q}, \boldsymbol{r})^2 = \prod_{l=1}^{L} \sum_{a_l=1}^{A} \left[ \frac{1}{2} \boldsymbol{r}[a_l, l]\boldsymbol{p}[a_l, l] + \frac{1}{2} \boldsymbol{r}[a_l, l]\boldsymbol{q}[a_l, l] \right] - \prod_{l=1}^{L} \sum_{a_l=1}^{A} \boldsymbol{r}[a_l, l]\sqrt{\boldsymbol{p}[a_l, l]\boldsymbol{q}[a_l, l]}$$

$$= \prod_{l=1}^{L} \sum_{a_l=1}^{A} \frac{1}{2} \mathbb{E}_p[\boldsymbol{x}]\boldsymbol{r}[a_l, l] + \frac{1}{2} \mathbb{E}_q[\boldsymbol{x}]\boldsymbol{r}[a_l, l] - \prod_{l=1}^{L} \sum_{a_l=1}^{A} \boldsymbol{r}[a_l, l]\sqrt{\boldsymbol{p}[a_l, l]\boldsymbol{q}[a_l, l]}.$$

*Proof.*

$$\text{HD}_{\boldsymbol{r}}(\boldsymbol{p}, \boldsymbol{q})^2 = \frac{1}{2} \sum_{\boldsymbol{x} \in \mathbb{X}} r(\boldsymbol{x}) \left( \sqrt{\boldsymbol{p}(\boldsymbol{x})} - \sqrt{\boldsymbol{q}(\boldsymbol{x})} \right)^2 \tag{29}$$

$$= \frac{1}{2} \sum_{\boldsymbol{x} \in \mathbb{X}} r(\boldsymbol{x}) \left( \boldsymbol{p}(\boldsymbol{x}) - 2\sqrt{\boldsymbol{p}(x)\boldsymbol{q}(x)} + q(\boldsymbol{x}) \right) \tag{30}$$

$/\!\!/$ *expanding the square*

$$= \frac{1}{2} \sum_{\boldsymbol{x} \in \mathbb{X}} \left[ \boldsymbol{r}(\boldsymbol{x})\boldsymbol{p}(\boldsymbol{x}) - 2\boldsymbol{r}(\boldsymbol{x})\sqrt{\boldsymbol{p}(\boldsymbol{x})\boldsymbol{q}(\boldsymbol{x})} + \boldsymbol{r}(\boldsymbol{x})\boldsymbol{q}(\boldsymbol{x}) \right] \tag{31}$$

$/\!\!/$ *note that* $\displaystyle\sum_{\boldsymbol{x} \in \mathbb{X}} \boldsymbol{r}(\boldsymbol{x})\boldsymbol{p}(\boldsymbol{x}) \neq 1$

$$= \frac{1}{2} \sum_{\boldsymbol{x} \in \mathbb{X}} \boldsymbol{r}(\boldsymbol{x})\boldsymbol{p}(\boldsymbol{x}) + \frac{1}{2} \sum_{\boldsymbol{x} \in \mathbb{X}} \boldsymbol{r}(\boldsymbol{x})\boldsymbol{q}(\boldsymbol{x}) - \sum_{\boldsymbol{x} \in \mathbb{X}} \boldsymbol{r}(\boldsymbol{x})\sqrt{\boldsymbol{p}(\boldsymbol{x})\boldsymbol{q}(\boldsymbol{x})} \tag{32}$$

$/\!\!/$ *property of the Hellinger distance*

$$= \frac{1}{2} \underbrace{\sum_{a_1=1}^{A} \cdots \sum_{a_L=1}^{A}}_{L \text{ times}} \boldsymbol{r}(x_1, \ldots, x_L)\boldsymbol{p}(x_1, \ldots, x_L) + \frac{1}{2} \underbrace{\sum_{a_1=1}^{A} \cdots \sum_{a_L=1}^{A}}_{L \text{ times}} \boldsymbol{r}(x_1, \ldots, x_L)\boldsymbol{q}(x_1, \ldots, x_L) \tag{33}$$

$$- \underbrace{\sum_{a_1=1}^{A} \cdots \sum_{a_L=1}^{A}}_{L \text{ times}} \boldsymbol{r}(x_1, \ldots, x_L)\sqrt{\boldsymbol{p}(x_1, \ldots, x_L)\boldsymbol{q}(x_1, \ldots, x_L)} \tag{34}$$

$/\!\!/$ *factorize*

$$= \frac{1}{2} \underbrace{\sum_{a_1=1}^{A} \cdots \sum_{a_L=1}^{A}}_{L \text{ times}} \prod_{l=1}^{L} \boldsymbol{r}[a_l, l]\boldsymbol{p}[a_l, l] + \frac{1}{2} \underbrace{\sum_{a_1=1}^{A} \cdots \sum_{a_L=1}^{A}}_{L \text{ times}} \prod_{l=1}^{L} \boldsymbol{r}[a_l, l]\boldsymbol{q}[a_l, l] \tag{35}$$

$$- \underbrace{\sum_{a_1=1}^{A} \cdots \sum_{a_L=1}^{A}}_{L \text{ times}} \prod_{l=1}^{L} \boldsymbol{r}[a_l, l]\sqrt{\boldsymbol{p}[a_l, l]\boldsymbol{q}[a_l, l]} \tag{36}$$

$/\!\!/$ *rearrange, and sums of products as products of sums - see C*

$$= \prod_{l=1}^{L} \sum_{a_l=1}^{A} \frac{1}{2}\boldsymbol{r}[a_l, l]\boldsymbol{p}[a_l, l] + \frac{1}{2}\boldsymbol{r}[a_l, l]\boldsymbol{q}[a_l, l] - \prod_{l=1}^{L} \sum_{a_l=1}^{A} \boldsymbol{r}[a_l, l]\sqrt{\boldsymbol{p}[a_l, l]\boldsymbol{q}[a_l, l]} \tag{37}$$

$\square$

## D.2 Hidden Markov model weighting

If $\boldsymbol{p}$ and $\boldsymbol{w}$ are both hidden Markov models, $k_w(\boldsymbol{p}, \mathbf{1}_x)$ can be evaluated efficiently. In this work, the setup is even simpler as we only consider Dirac distributions for $\boldsymbol{p}$ (see Section 4.2). In that case, for $\boldsymbol{x} \neq \boldsymbol{x}'$, $\text{HD}_{\boldsymbol{w}}(\mathbb{1}_x, \mathbb{1}_{x'}) = \sqrt{\frac{\boldsymbol{w}(\boldsymbol{x}) + \boldsymbol{w}(\boldsymbol{x}')}{2}}$ where $w(\boldsymbol{x})$ is computed by the forward algorithm (see for example Bishop (2006, Chapter 13.2)). We obtain our weightings by running HMMER (Durbin et al., 1998; Potter et al., 2018) with default parameters on the wild-type and all given unlabeled sequences. Our code repository contains a shell-script to do this.

### D.3 PLM weighting

We obtain $\boldsymbol{p}$ and $\boldsymbol{w}$ from a PLM by the likelihoods from a softmax on the last-layer logits of e.g.~esm2, where the input sequence is masked at every position as described in (Rives et al., 2021) and the implementation of Notin et al. (2023) (see `masked-marginals` in `proteingym/baselines/esm/compute_fitness`).

### D.4 Stabilizing the product kernel weightings

The product kernel presented in Sec. 3.2.1 can yield numerical underflow, i.e., when the number of kernel components is very large or the individual weights become vanishingly small. Under those conditions the computation can be stabilized by computing covariance function values and weightings in log-space instead.

## E Evaluations of f and the argmax p on acquisition

The function $f$ does not act on the space of probability measures and there is no bijective mapping between a probability vector and a discrete $\boldsymbol{x}$. Ideally, the optimization of $\alpha$ arrives at a Dirac distribution, meaning $\boldsymbol{p}$ is of the form $\mathbb{1}_x$. Then mapping the optimization outcome to a sequence is unambiguous. In the other case, when $\boldsymbol{p}$ is not a Dirac, we can sample sequences from $\boldsymbol{p}$, and pick the $\boldsymbol{x}$ for evaluation which has the best value of $\alpha(\mathbb{1}_x)$.

## F `CoRel` algorithm specifications

### F.1 A continuous optimization algorithm

---
**Algorithm 3** `CoRel` using continuous optimization
---
**Input:** acquisition $a : \mathbb{P}_f \to \mathbb{R}$, black-box $f : \mathbb{X} \to \mathbb{R}$, dataset $\mathcal{D}_1 = \{X, y\}$, pretrained LVM $\phi : \mathbb{R}^D \to \mathbb{P}_f$
   **for** $t \in 1, ..., t_{\max}$ **do**
      $m \leftarrow \text{trainModel}((\mathbb{1}_{\boldsymbol{x}_i}, \boldsymbol{y}_i)_{i=1}^t)$
      $\boldsymbol{p}_* \leftarrow \arg\max_{\boldsymbol{p}} a(\boldsymbol{p}, m)$
      $\boldsymbol{x} \leftarrow \text{getSequenceFromDistribution}(\boldsymbol{p}_*)$
      $\mathcal{D}_{t+1} \leftarrow \mathcal{D}_t \cup \{\boldsymbol{x}, f(\boldsymbol{x})\}$
   **end for**
---

### F.2 A discrete optimization algorithm

---
**Algorithm 4** `CoRel` using discrete optimization
---
**Input:** acquisition $a : \mathbb{P}_f \to \mathbb{R}$, black-box $f : \mathbb{X} \to \mathbb{R}$, dataset $\mathcal{D}_1 = \{X, y\}$, pretrained LVM $\phi : \mathbb{R}^D \to \mathbb{P}_f$
   **for** $t \in 1, ..., t_{\max}$ **do**
      $m \leftarrow \text{trainModel}((\mathbb{1}_{\boldsymbol{x}_i}, \boldsymbol{y}_i)_{i=1}^t)$
      $\boldsymbol{x} \leftarrow \arg\max_{\boldsymbol{x}'} a(\mathbb{1}_{\boldsymbol{x}'}, m)$
      $\mathcal{D}_{t+1} \leftarrow \mathcal{D}_t \cup \{\boldsymbol{x}, f(\boldsymbol{x})\}$
   **end for**
---

### F.3 Optimizing multiple properties

Given an optimization task for multiple properties (see RFP optimization), we require a function for finding Pareto optimal points. Given a set of all points $S$ with $x \subset S$:

$$p_{\text{opt}}(x) := \{x \in S | \nexists x' \in S \text{ s.t. } x' \preceq x \wedge x' \neq x\}. \tag{38}$$

In our experiments pareto optimal points are determined by the $y$ vector.

## G  Convergence and regret considerations

### G.1  Discrete convergence

Specifically, assume $\alpha$ is UCB and given the corners of the constraint simplex $\{\delta_x\}$, which is of size $|A|^L$, then every evaluation of $f$ takes a discrete sequence $\boldsymbol{x}$ which is a finite-arm bandit with $|A|^L$ arms – one for each corner. Each acquisition at step $t$ now with UCB variance-scaling parameter $\beta$ we select

$$\boldsymbol{p}_t = \arg\max_{\boldsymbol{p} \in \delta_x}[\mu_{t-1}(\boldsymbol{p}) + \beta_t^{\frac{1}{2}}\sigma_{t-1}(\boldsymbol{p})]. \tag{39}$$

**Proposition 6.** *Assume $\alpha$ is UCB, then optimizing $\bar{f}$ such that only $\delta_{\boldsymbol{x}}$ are evaluated yields sub-linear regret.*

*Proof sketch.* Let the search be over the finite set $\{\delta_x : x \in A^L\}$ and place a GP prior on $\bar{f}$ with any p.d. kernel $k$, then any standard finite-arm GP-bandit algorithm applies (*i.e.* UCB, TS) if the candidate set is restricted to $\{\delta_x\}$. We use the finite-arm regret theorem under additional assumptions in Srinivas et al. (2012) such that the final regret bound $\mathcal{R}_T \leq O(\sqrt{T\beta_T\gamma_T})$ with high probability; where $\gamma_T$ is an upper bounded maximum gain in information. $\square$

### G.2  Continuous convergence

Performing continuous optimization directly on $\mathbb{P}_f$ or—given a probabilistic decoder—in the latent space that maps onto $\mathbb{P}_f$ does not give standard continuous BO convergence. The required results apply only under a set of assumptions on the model, and its evaluations (Jones et al., 1998; Garnett, 2022). Specifically, $f$ is required to belong to the kernel RKHS for GP-UCB to converge, however our $f$ is a function on the discrete realized (Dirac) input sequences, whereas the surrogate model and subsequent RKHS are formulated on the continuous $\mathbb{P}_f$. Strictly speaking, the model is thus misspecified, prohibiting the use of continuous convergence results. Other convergence requirements are a compact $\mathbb{P}_f$, any set of latents to be compact, and $k$ to be Lipschitz continuous. To address the $f$ RKHS mismatch we need to consider to what extent $f$ is *approximately* in the surrogate $\bar{f}$ RKHS with the proposed kernel such that the results by Bogunovic and Krause (2021) on EC-GP-UCB can apply.

## H  Experimental Setup

| Identifier | Type | Dimensions | Task | Reference |
|---|---|---|---|---|
| RFP | amino acids | $20^{396}$ | stability & surface accessibility (2D) | Stanton et al. (2022) |
| GFP | amino acids | $20^{237}$ | CBas value (1D) | Brookes et al. (2019) |
| PMO | selfie tokens | $64^{70}$ | molecular properties (1D) | Gao et al. (2022) |

Table A2: Experiment Overview

## I  Baseline implementations and hyperparameters

### I.1  Implementation and optimization

Models and experiments are implemented with Tensorflow (Apache License 2.0) (Martín Abadi et al., 2015) (`tf`), Tensorflow-probability (Apache License 2.0), GPflow (Apache License 2.0) (Matthews et al., 2017), and Trieste (MIT license) (Picheny et al., 2023). We provide a vectorized implementation of the weighted hellinger kernel and base hellinger kernel in TF under the MIT license. Model hyperparameters are optimized using the scipy LBFGS optimizer (BSD 3-Clause License) (Virtanen et al., 2020) on the model likelihood as previously described. All results have been recorded with MlFlow (Apache License 2.0), and `wandb` (MIT license).

### I.2 Computational Resources

Initial development was done on a M1 Pro ARM architecture with tensorflow-metal support on a MacOS. All final experiments have been run on a Linux HPC platform (4.18.0) x86_64 architecture with Intel Xeon 6248 CPUs. GPU resources include NVIDIA Titan Xp, RTX, Quadro, A40 with CUDA version 12.3.

### I.3 Discrete biological sequence optimization library

This library contains the RFP, GFP, and all PMO problems, as well as the stable *LamBO* implementation for experimental queries (Apache License 2.0). We define the RFP problem with FoldX (Academic License) and SASA computations (BSD 3-Clause License of RDKit (Landrum, 2024)), respective the LamBO defined pareto front. We additionally include a reference objective that is equivalent to the LamBO setup and includes additional sequences.

### I.4 *CBas* VAE model

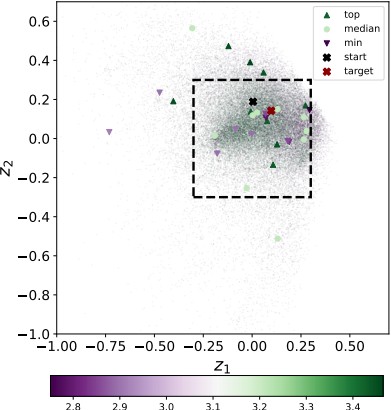

Figure A1: The latent space ($z \in \mathbb{R}^2$), encodes the corpus of all experimentally evaluated sequences (dots). Available for optimization are only oracle evaluations - see markers (top ▲, median ●, and lowest ▼ 10 observations each). Start is wild-type and target the maximally fluorescent candidate. Fig. 1 displays the dashed square latent space region.

The *GFP* problem is presented in Brookes et al. (2019) with a VAE as a latent variable model and a custom predictive GP model.[8] The full training corpus are 54025 unique sequences for which observations are available and is originally provided in Sarkisyan et al. (2016). For the VAE: the encoder is a simple neural network (size=50 units) with ELU activation mapping to 20 latent dimensions. The decoder is a deep neural network with 3 layers of dimensions [50, 20*len(sequence), 20] with ELU and softmax activation respectively, such that we first obtain a mapping from latent space to hidden space of size number of amino acids times (aligned) sequence length and ultimately the label-encoded protein sequence. Model input are aligned GFP sequences.

Training data are 5000 sequences of length 237 with 20 possible amino-acid tokens at each position. The oracle training data are the upper-quantile (by fluorescence measurement), which is approximately 9% of all available sequences. Training has been done for 100 epochs in batches of 10 using a (tensorflow2) Adam optimizer with default configuration.

The model latent space is used in combination with a predictive GP oracle model as a surrogate for *GFP*. For a specification of the predictive GP model we refer to Brookes et al. (2019).

---

[8]The authors provide **no** license, therefore we follow the ICML copyright by authors and PMLR 2023.

The optimization task for the GFP problem is the minimization of the negative oracle values, to find the global minimum - see Fig. 4. For later analysis the sign is inverted, as we ultimately intend to maximize the oracle.

We adapt this architecture to two latent dimensions for visualization purposes only. All empirical results queried against the GP utilize the initial model of full dimensionality. The remaining model specifications remain the same.

### I.5 LamBO latent optimizer

We use the LamBO implementation directly from the stable tagged submission branch of Stanton et al. (2022) (commit `22afec26da0b9ea1810e65f8a60ea7988c021cef` Oct-2022) and refer to their exact model specifications. We optimize the multi-objective RFP problem using NEHVI (Daulton et al., 2022) acquisition. The only addition we make is to define a dedicated task protocol to work with the discrete sequence optimization library we provide, and associated configurations.
We highlight that the original model specification and experiments account for a large range of starting sequences including seed-sequences. Additionally, when analyzing the model and observations we find that the relative hypervolume is computed respective a hard-coded set of reference values. For our benchmarking purposes we record this value also (see Supplementary Fig. A5). In our case we obtain the start values from the black-box function evaluations and compute the relative HV with respect to that. We provide the algorithm only with an exact set of 6 RFP PDB files to use, and do not access any provided seed-sequences.

The LamBO RFP data-set contains 50 PDB files (RFP structure files), 754 related sequences, and 1923 generated proxy seed sequences.

### I.6 PMO VAE

We optimize the PMO tasks in a learned latent space, via a standard VAE architecture trained on ZINC250k with `selfies` tokenization. Implementations of an identical architecture are provided in Tensorflow (for `CoRel`) and PyTorch (for all torch-based models). The encoder is a three layer architecture with 2048, 1024, 256 dense layers, to the latent dimensions ($\times 2$ for $\mu$ and $\sigma$ of the latent). All encoding layers have batch-normalization, and dropout ($p = 0.2$) for each layer, with a ReLU activation function. The `tf` implementation learns a `tf` Probability `MultivariateNormalDiag`, whereas the `torch` version learns a `Normal` distribution object. The decoder is symmetrical to the encoder, with latent dimensions to 256, 1024, and 2048 dense layers, each with batch-norm, activation, and dropout, as described in the encoder. The VAE is trained to minimize a negative ELBO with a discounted KL divergence: $-ELBO := -\mathbb{E}_{q(z|x)}p(x|z) + 0.1 * D_{KL}(q(z|x)||p(z))$ .

We train a $d = 128$ latent variable model (VAE) which is used for all presented algorithms. In an ablation we consider $d = 64$ latent dimensions (see Appendix J.3.1).

Model training for `tf` was stopped after 1050 iterations, with a train loss of 13.465 and test loss of 11.095, and `torch` obtained a train loss of 11.188 and test loss of 17.904 at the same number of iterations.

We highlight that the training-/validation-loss show discrepancies between the two which are likely to implementational differences on the library side (optimizer, sampling, etc.).

### I.7 Random Mutations

As a naïve baseline, we consider mutating positions at random in the Pareto front. At each iteration, we maintain a population of 16 elements (padding with random mutations if the Pareto front is not large enough). Retaining well performing mutations leads to a hill-climbing algorithm. Each of these elements is then mutated in two random positions (taking into account that sequences may have varying lengths, and never performing any inserting/deleting operations). The results provided were averaged over 5 iterations with different random seeds. The performance of this baseline is noteworthy, and the difference between it and e.g. Stanton et al. (2022)'s NSGA-2 may be attributed to the fact that we mutate *twice* instead of *once* per iteration.

### I.8 Probabilistic Reparameterization

We set up Probabilistic Reparameterization (`PR`) as presented in Daulton et al. (2022) and the codebase available with it. Specifically using EI for single objective optimization. We find that the high dimensionality of our problem space poses a challenge and we treat the problem therefore as a categorical problem space (for RFP, GFP) whereas the base implementation may convert to one-hot encoding of the problem (PMO). The GFP problem is implemented as a single task problem, whereas RFP as a multi-task problem. Conducting the GFP experiments was computationally feasible in the categorical setup, however the algorithm suggests a lot of potential mutations per iteration. Since the RFP task relies on running stability simulations via FoldX and the runtime of the method increases with the number of mutations proposed, we find that running the RFP experiments within the given budget to be prohibitively expensive. We remark that apart from high runtime of the black-box functions, the very high number of proposed mutations is an unrealistic scenario for protein engineering tasks.

### I.9 Sobol sampling

We use the Sobol sampler as provided with the Probabilistic Reparameterization codebase from Daulton et al. (2022). In contrast to the previously described *random mutations* (see Appendix I.7) the whole sequence *i.e.* every position is sampled completely at random. We remark that this algorithm yields a very large number of mutations compared to a reference sequence because potentially any position can be mutated to any other available amino acid. This is an unrealistic scenario for any protein engineering task and therefore serves only as a random baseline.

### I.10 NSGA-II

During the development and initial experimental runs we had included the Pymoo implementation of NSGA-II Blank and Deb (2020) for multi-objective optimization. However, due to implementation constraints we found that the NSGA-II algorithm proposes a very large number of mutations for candidates, making a direct comparison to `CoRel` and LamBO challenging. On one side, this makes the experimental runs significantly more expensive than `CoRel`, LamBO or random HC, while on the other hand this yields quite extreme task values which appear initially like a significant improvement. We remark that proposing more than 20 mutations per iteration is however quite challenging in a protein engineering setting. An adaptation of NSGA-II which includes a hard limit on number of mutations we leave for future work.

### I.11 CMA-ES

We find the CMA-ES algorithm performs best for a range of tasks in Table A3. This suggests that the latent variable model (VAE), used to optimize the PMO tasks, are sufficiently low in dimensionality and the covariance structure correlated with the downstream objective. Still, the dimensionality of the latent model is sufficiently large to pose a challenge the BO baseline methods (e.g. PR, Bounce, VanillaBO).

### I.12 PMO optimizers

For the PMO comparisons, we follow the implementations in González-Duque et al. (2024) exactly. Which is PR and Bounce using Expected Improvement, Turbo by default uses Thompson sampling, VanillaBO (following (Hvarfner et al., 2023)) using log-noisy Expected Improvement.

## J Additional Results

### J.1 Reported Metrics

$$\text{std.err.} = \text{SE} := \frac{\sigma}{\sqrt{n}}$$

$$\text{std.dev.} = \text{SD} = \sigma := \sqrt{\sum_{i=1}^{N} \frac{(x_i - \mu)^2}{N}}$$

$$\mu = \frac{1}{N} \sum_{i=1}^{N} x_i$$

$$\mu_{\text{norm}} = \frac{1}{N} \sum_{i=1}^{N} \frac{x_i}{x_{\max}}$$

In Table A3 we report the mean of the best observed values across seeds and standard deviation ($\pm$) across seeds. In Table 1 we normalize each task (row) in Table A3 by the max value, such that the best observed value per task is 1., and compute the empirical mean across the reported normalized value and standard deviation.

## J.2  PMO

Table A3: PMO results, average best function values over 9 seeds with standard deviation across seeds ($\pm$), budget is 300 black-box evaluations.

| group | oracle | BO CoRel | PR | Bounce | Turbo | VanillaBO | ref. CMA-ES | random HC |
|---|---|---|---|---|---|---|---|---|
| optimize | rdkit_logp | $7.49 \pm 0.95$ | $5.23 \pm 0.75$ | $6.50 \pm 3.38$ | $\mathbf{10.57} \pm 1.61$ | $7.86 \pm 0.82$ | $9.88 \pm 1.27$ | $6.89 \pm 0.25$ |
|  | rdkit_qed | $\mathbf{0.94} \pm 0.0$ | $0.80 \pm 0.06$ | $0.53 \pm 0.13$ | $0.89 \pm 0.05$ | $0.88 \pm 0.04$ | $0.90 \pm 0.02$ | $0.88 \pm 0.01$ |
|  | sa_tdc | $8.78 \pm 0.13$ | $\mathbf{8.93} \pm 0.10$ | $8.79 \pm 0.16$ | $7.70 \pm 0.24$ | $7.60 \pm 0.26$ | $8.02 \pm 0.14$ | $7.95 \pm 0.05$ |
| dock | drd2_docking | $0.03 \pm 0.01$ | $0.02 \pm 0.01$ | $0.01 \pm 0.01$ | $0.14 \pm 0.02$ | $0.13 \pm 0.08$ | $\mathbf{0.20} \pm 0.13$ | $0.16 \pm 0.13$ |
|  | gsk3_beta | $\mathbf{0.2} \pm 0.04$ | $0.00 \pm 0.00$ | $0.00 \pm 0.00$ | $0.00 \pm 0.00$ | $0.00 \pm 0.00$ | $0.00 \pm 0.00$ | $0.00 \pm 0.00$ |
|  | jnk3 | $0.09 \pm 0.02$ | $0.08 \pm 0.01$ | $0.08 \pm 0.08$ | $\mathbf{0.14} \pm 0.06$ | $0.10 \pm 0.03$ | $0.11 \pm 0.01$ | $0.08 \pm 0.02$ |
| discover | celecoxib_rediscovery | $0.21 \pm 0.0$ | $0.13 \pm 0.02$ | $0.02 \pm 0.01$ | $\mathbf{0.22} \pm 0.04$ | $0.21 \pm 0.01$ | $\mathbf{0.22} \pm 0.03$ | $0.18 \pm 0.03$ |
|  | thiothixene_rediscovery | $0.22 \pm 0.0$ | $0.17 \pm 0.03$ | $0.02 \pm 0.01$ | $\mathbf{0.24} \pm 0.01$ | $0.22 \pm 0.02$ | $0.23 \pm 0.02$ | $0.23 \pm 0.01$ |
|  | troglitazone_rediscovery | $0.16 \pm 0.01$ | $0.11 \pm 0.02$ | $0.02 \pm 0.00$ | $0.20 \pm 0.03$ | $0.18 \pm 0.01$ | $\mathbf{0.21} \pm 0.01$ | $0.16 \pm 0.00$ |
| mpo | amlodipine_mpo | $0.44 \pm 0.0$ | $0.24 \pm 0.10$ | $0.00 \pm 0.00$ | $0.45 \pm 0.04$ | $0.45 \pm 0.01$ | $\mathbf{0.48} \pm 0.05$ | $0.45 \pm 0.03$ |
|  | fexofenadine_mpo | $0.54 \pm 0.12$ | $0.36 \pm 0.17$ | $0.25 \pm 0.00$ | $\mathbf{0.70} \pm 0.05$ | $0.65 \pm 0.03$ | $0.64 \pm 0.02$ | $0.63 \pm 0.02$ |
|  | osimetrinib_mpo | $0.63 \pm 0.03$ | $0.60 \pm 0.02$ | $0.58 \pm 0.05$ | $0.72 \pm 0.01$ | $0.70 \pm 0.02$ | $\mathbf{0.74} \pm 0.02$ | $0.70 \pm 0.05$ |
|  | perindopril_mpo | $0.32 \pm 0.0$ | $0.10 \pm 0.11$ | $0.00 \pm 0.00$ | $0.33 \pm 0.09$ | $0.28 \pm 0.03$ | $\mathbf{0.37} \pm 0.06$ | $0.26 \pm 0.09$ |
|  | ranolazine_mpo | $0.34 \pm 0.07$ | $0.11 \pm 0.04$ | $0.13 \pm 0.22$ | $\mathbf{0.68} \pm 0.08$ | $0.43 \pm 0.08$ | $0.63 \pm 0.08$ | $0.37 \pm 0.17$ |
|  | sitagliptin_mpo | $0.08 \pm 0.07$ | $0.03 \pm 0.02$ | $0.00 \pm 0.00$ | $0.26 \pm 0.02$ | $0.28 \pm 0.04$ | $\mathbf{0.35} \pm 0.10$ | $0.23 \pm 0.13$ |
|  | zaleplon_mpo | $0.21 \pm 0.02$ | $0.09 \pm 0.03$ | $0.00 \pm 0.00$ | $0.37 \pm 0.04$ | $0.33 \pm 0.01$ | $\mathbf{0.40} \pm 0.02$ | $0.34 \pm 0.05$ |
| other | albuterol_similarity | $0.33 \pm 0.03$ | $0.30 \pm 0.08$ | $0.17 \pm 0.03$ | $\mathbf{0.52} \pm 0.06$ | $0.46 \pm 0.06$ | $0.44 \pm 0.05$ | $0.46 \pm 0.09$ |
|  | deco_hop | $0.53 \pm 0.0$ | $0.52 \pm 0.01$ | $0.51 \pm 0.01$ | $\mathbf{0.56} \pm 0.02$ | $\mathbf{0.56} \pm 0.00$ | $0.55 \pm 0.01$ | $0.54 \pm 0.02$ |
|  | isomer_c7h8n2o2 | $0.54 \pm 0.08$ | $0.33 \pm 0.18$ | $0.24 \pm 0.31$ | $0.87 \pm 0.13$ | $0.77 \pm 0.16$ | $\mathbf{0.92} \pm 0.07$ | $0.82 \pm 0.08$ |
|  | isomer_c9h10n2o2pf2cl | $0.37 \pm 0.14$ | $0.21 \pm 0.09$ | $0.05 \pm 0.08$ | $0.64 \pm 0.08$ | $0.63 \pm 0.06$ | $\mathbf{0.76} \pm 0.07$ | $0.51 \pm 0.09$ |
|  | median_1 | $0.13 \pm 0.01$ | $0.13 \pm ?$ | $0.01 \pm 0.01$ | $0.15 \pm 0.01$ | $\mathbf{0.19} \pm 0.04$ | $\mathbf{0.19} \pm 0.01$ | $\mathbf{0.19} \pm 0.05$ |
|  | median_2 | $0.12 \pm 0.0$ | $0.09 \pm 0.00$ | $0.01 \pm 0.00$ | $\mathbf{0.15} \pm 0.01$ | $0.14 \pm 0.01$ | $\mathbf{0.15} \pm 0.01$ | $\mathbf{0.15} \pm 0.01$ |
|  | mestranol_similarity | $0.31 \pm 0.03$ | $0.29 \pm 0.02$ | $0.03 \pm 0.01$ | $\mathbf{0.44} \pm 0.06$ | $0.36 \pm 0.02$ | $0.41 \pm 0.03$ | $0.37 \pm 0.02$ |
|  | scaffold_hop | $0.38 \pm 0.0$ | $0.36 \pm 0.01$ | $0.35 \pm 0.01$ | $0.39 \pm 0.01$ | $0.40 \pm 0.02$ | $\mathbf{0.44} \pm 0.02$ | $0.39 \pm 0.02$ |
|  | valsartan_smarts | $0.0 \pm 0.0$ | $0.00 \pm 0.00$ | $0.00 \pm 0.00$ | $0.00 \pm 0.00$ | $0.00 \pm 0.00$ | $0.00 \pm 0.00$ | $0.00 \pm 0.00$ |

`PR` and `Bounce` fail due to memory requirements, and terminate early. For these methods all values up until the point of failure are recorded and displayed in results. Less runs fail for the lower-dimensional VAE ablation ($d = 64$) for `PR` and `Bounce`.

## J.3 Ablation: Product kernel of different sizes (PMO VAE d=128)

| group | CoRel (N=250) value | ±SD | CoRel (N=2.5k) value | ±SD | CoRel (N=25k) value | ±SD | BO PR value | ±SD | Bounce value | ±SD | Turbo value | ±SD | VanillaBO value | ±SD | ref. CMA-ES value | ±SD | random HC value | ±SD |
|---|---|---|---|---|---|---|---|---|---|---|---|---|---|---|---|---|---|---|
| optimize | 0.90 | 0.03 | 0.86 | 0.06 | 0.62 | 0.00 | 0.78 | 0.05 | 0.72 | 0.16 | 0.94 | 0.08 | 0.84 | 0.05 | 0.93 | 0.05 | 0.83 | 0.01 |
| discover | 0.86 | 0.00 | 0.89 | 0.03 | 0.86 | 0.00 | 0.61 | 0.10 | 0.09 | 0.03 | 0.98 | 0.12 | 0.91 | 0.06 | 0.99 | 0.09 | 0.85 | 0.09 |
| dock | 0.61 | 0.17 | 0.53 | 0.16 | 0.39 | 0.00 | 0.22 | 0.04 | 0.21 | 0.21 | 0.57 | 0.18 | 0.45 | 0.20 | 0.60 | 0.24 | 0.46 | 0.26 |
| mpo | 0.68 | 0.07 | 0.60 | 0.12 | 0.39 | 0.00 | 0.37 | 0.14 | 0.19 | 0.06 | 0.92 | 0.10 | 0.83 | 0.08 | 0.98 | 0.11 | 0.79 | 0.16 |
| other | 0.62 | 0.05 | 0.59 | 0.07 | 0.47 | 0.00 | 0.54 | - | 0.28 | 0.07 | 0.83 | 0.08 | 0.80 | 0.08 | 0.86 | 0.06 | 0.79 | 0.09 |

Table A4: Aggregated results with d=128 VAE, evaluating `CoRel` with different kernel-sample sizes (N=250,2500,25.000) (default N=1000) (5 seeds, 9 seeds for baselines). We find that product kernels constructed with more samples do not necessarily yield performance increases.

## J.3.1 Ablation: PMO VAE d=64

| group | CoRel (N=1k) value | ±SD | CoRel (N=25k) value | ±SD | CoRel (N=2.5k) value | ±SD | BO PR value | ±SD | Bounce value | ±SD | Turbo value | ±SD | VanillaBO value | ±SD | ref. CMA-ES value | ±SD | random HC value | ±SD |
|---|---|---|---|---|---|---|---|---|---|---|---|---|---|---|---|---|---|---|
| optimize | 0.87 | 0.01 | 0.66 | 0.00 | 0.86 | 0.01 | 0.80 | 0.10 | 0.67 | 0.12 | 0.92 | 0.11 | 0.89 | 0.04 | 0.92 | 0.06 | 0.91 | 0.07 |
| discover | 0.90 | 0.00 | 0.90 | 0.00 | 0.90 | 0.00 | 0.56 | 0.11 | 0.08 | 0.05 | 0.89 | 0.03 | 0.78 | 0.08 | 0.92 | 0.11 | 0.91 | 0.09 |
| dock | 0.40 | 0.10 | 0.30 | 0.00 | 0.40 | 0.11 | 0.46 | 0.22 | 0.38 | 0.40 | 0.76 | 0.15 | 0.69 | 0.13 | 0.63 | 0.17 | 0.92 | 0.56 |
| mpo | 0.67 | 0.05 | 0.52 | 0.10 | 0.67 | 0.06 | 0.47 | 0.17 | 0.16 | 0.13 | 0.89 | 0.13 | 0.87 | 0.14 | 0.94 | 0.13 | 0.88 | 0.16 |
| other | 0.65 | 0.08 | 0.55 | 0.04 | 0.64 | 0.08 | 0.60 | - | 0.29 | 0.04 | 0.81 | 0.05 | 0.84 | 0.10 | 0.80 | 0.07 | 0.82 | 0.10 |

Table A5: Aggregated results with d=64 VAE, evaluating `CoRel` with three different kernel-sample sizes (5 seeds `CoRel`, 9 seeds for remainder) for 100 iterations. We find that the lower dimensions are beneficial for `PR`, `VanillaBO`, and the `random HC` baseline. Unaggregated results are in Table A6

| group | oracle | CoRel (N=1k) | CoRel (N=25k) | CoRel (N=2.5k) | BO PR | Bounce | Turbo | VanillaBO | ref. CMA-ES | random HC |
|---|---|---|---|---|---|---|---|---|---|---|
| optimize | rdkit_logp | 5.22 ± 0.3 | 5.05 ± 0.0 | 5.1 ± 0.09 | 5.95 ± 1.25 | 4.85 ± 2.42 | 8.53 ± 2.20 | 7.35 ± 0.50 | 7.89 ± 1.02 | 8.21 ± 0.92 |
| | rdkit_qed | 0.94 ± 0.0 | 0.94 ± 0.0 | 0.94 ± 0.0 | 0.67 ± 0.13 | 0.42 ± 0.03 | 0.84 ± 0.03 | 0.87 ± 0.04 | 0.89 ± 0.03 | 0.87 ± 0.03 |
| | sa_tdc | 8.57 ± 0.07 | 3.43 ± 0.0 | 8.55 ± 0.06 | 8.70 ± 0.12 | 8.67 ± 0.30 | 7.65 ± 0.30 | 7.79 ± 0.07 | 7.74 ± 0.24 | 7.44 ± 0.56 |
| dock | drd2_docking | 0.02 ± 0.0 | 0.02 ± 0.0 | 0.02 ± 0.01 | 0.02 ± 0.00 | 0.01 ± 0.01 | 0.08 ± 0.04 | 0.04 ± 0.02 | 0.06 ± 0.02 | 0.18 ± 0.24 |
| | gsk3_beta | 0.16 ± 0.0 | 0.16 ± 0.0 | 0.16 ± 0.0 | 0.22 ± 0.11 | 0.12 ± 0.11 | 0.29 ± 0.06 | 0.28 ± 0.06 | 0.23 ± 0.10 | 0.31 ± 0.12 |
| | jnk3 | 0.06 ± 0.03 | 0.03 ± 0.0 | 0.06 ± 0.03 | 0.06 ± 0.03 | 0.07 ± 0.08 | 0.10 ± 0.01 | 0.10 ± 0.01 | 0.09 ± 0.01 | 0.10 ± 0.03 |
| discover | celecoxib_rediscovery | 0.21 ± 0.0 | 0.21 ± 0.0 | 0.21 ± 0.0 | 0.12 ± 0.04 | 0.02 ± 0.01 | 0.20 ± 0.00 | 0.15 ± 0.01 | 0.19 ± 0.03 | 0.17 ± 0.02 |
| | thiothixene_rediscovery | 0.22 ± 0.0 | 0.22 ± 0.0 | 0.22 ± 0.0 | 0.12 ± 0.02 | 0.01 ± 0.01 | 0.20 ± 0.01 | 0.19 ± 0.03 | 0.22 ± 0.02 | 0.20 ± 0.03 |
| | troglitazone_rediscovery | 0.15 ± 0.0 | 0.15 ± 0.0 | 0.15 ± 0.0 | 0.12 ± 0.01 | 0.02 ± 0.01 | 0.17 ± 0.01 | 0.16 ± 0.01 | 0.18 ± 0.02 | 0.21 ± 0.01 |
| mpo | amlodipine_mpo | 0.44 ± 0.0 | 0.44 ± 0.0 | 0.44 ± 0.0 | 0.14 ± 0.14 | 0.00 ± 0.00 | 0.42 ± 0.01 | 0.42 ± 0.05 | 0.42 ± 0.03 | 0.42 ± 0.04 |
| | fexofenadine_mpo | 0.35 ± 0.09 | 0.15 ± 0.17 | 0.35 ± 0.08 | 0.58 ± 0.02 | 0.14 ± 0.13 | 0.61 ± 0.03 | 0.60 ± 0.01 | 0.62 ± 0.02 | 0.64 ± 0.03 |
| | osimetrinib_mpo | 0.61 ± 0.01 | 0.26 ± 0.28 | 0.61 ± 0.01 | 0.61 ± 0.03 | 0.42 ± 0.32 | 0.68 ± 0.02 | 0.67 ± 0.04 | 0.67 ± 0.03 | 0.68 ± 0.04 |
| | perindopril_mpo | 0.32 ± 0.0 | 0.32 ± 0.0 | 0.32 ± 0.0 | 0.07 ± 0.05 | 0.00 ± 0.00 | 0.31 ± 0.06 | 0.19 ± 0.06 | 0.30 ± 0.02 | 0.31 ± 0.06 |
| | ranolazine_mpo | 0.23 ± 0.0 | 0.23 ± 0.0 | 0.23 ± 0.0 | 0.18 ± 0.06 | 0.14 ± 0.14 | 0.40 ± 0.16 | 0.41 ± 0.03 | 0.51 ± 0.14 | 0.54 ± 0.11 |
| | sitagliptin_mpo | 0.06 ± 0.07 | 0.0 ± 0.0 | 0.06 ± 0.08 | 0.10 ± 0.09 | 0.00 ± 0.00 | 0.19 ± 0.05 | 0.32 ± 0.17 | 0.31 ± 0.09 | 0.18 ± 0.08 |
| | zaleplon_mpo | 0.2 ± 0.0 | 0.2 ± 0.0 | 0.2 ± 0.0 | 0.10 ± 0.09 | 0.00 ± 0.00 | 0.33 ± 0.06 | 0.28 ± 0.01 | 0.26 ± 0.05 | 0.22 ± 0.09 |
| other | albuterol_similarity | 0.31 ± 0.0 | 0.31 ± 0.0 | 0.31 ± 0.0 | 0.24 ± 0.03 | 0.17 ± 0.03 | 0.38 ± 0.01 | 0.42 ± 0.08 | 0.38 ± 0.06 | 0.38 ± 0.07 |
| | deco_hop | 0.53 ± 0.0 | 0.53 ± 0.0 | 0.53 ± 0.0 | 0.52 ± 0.00 | 0.51 ± 0.01 | 0.55 ± 0.01 | 0.53 ± 0.01 | 0.55 ± 0.02 | 0.54 ± 0.01 |
| | isomer_c7h8n2o2 | 0.37 ± 0.21 | 0.04 ± 0.09 | 0.34 ± 0.21 | 0.38 ± 0.29 | 0.08 ± 0.08 | 0.65 ± 0.10 | 0.87 ± 0.11 | 0.51 ± 0.11 | 0.70 ± 0.19 |
| | isomer_c9h10n2o2pf2cl | 0.28 ± 0.19 | 0.05 ± 0.11 | 0.23 ± 0.17 | 0.28 ± 0.14 | 0.01 ± 0.00 | 0.51 ± 0.02 | 0.55 ± 0.09 | 0.57 ± 0.01 | 0.43 ± 0.14 |
| | median_1 | 0.13 ± 0.01 | 0.12 ± 0.0 | 0.13 ± 0.01 | 0.11 ± 0.02 | 0.02 ± 0.02 | 0.14 ± 0.01 | 0.13 ± 0.03 | 0.15 ± 0.02 | 0.14 ± 0.01 |
| | median_2 | 0.12 ± 0.0 | 0.12 ± 0.0 | 0.12 ± 0.0 | 0.08 ± 0.02 | 0.01 ± 0.00 | 0.13 ± 0.01 | 0.13 ± 0.02 | 0.12 ± 0.01 | 0.14 ± 0.01 |
| | mestranol_similarity | 0.21 ± 0.03 | 0.2 ± 0.02 | 0.22 ± 0.03 | 0.30 ± 0.00 | 0.02 ± 0.01 | 0.33 ± 0.02 | 0.33 ± 0.02 | 0.34 ± 0.03 | 0.39 ± 0.04 |
| | scaffold_hop | 0.38 ± 0.0 | 0.38 ± 0.0 | 0.38 ± 0.0 | 0.36 ± 0.00 | 0.35 ± 0.01 | 0.39 ± 0.01 | 0.38 ± 0.01 | 0.39 ± 0.00 | 0.39 ± 0.02 |
| | valsartan_smarts | 0.0 ± 0.0 | 0.0 ± 0.0 | 0.0 ± 0.0 | 0.00 ± 0.00 | 0.00 ± 0.00 | 0.00 ± 0.00 | 0.00 ± 0.00 | 0.00 ± 0.00 | 0.00 ± 0.00 |

Table A6: PMO VAE d=64 optimization results. All methods have been assessed on a VAE with 64 latent dimensions, `CoRel` 5 seeds, remainder 9 seeds. Mean best observed values over 100 iterations (budget) across seeds and standard deviation are reported. We report for three different `CoRel` kernel methods, where product kernels have been sampled from different sample-sizes from ZINC250k (N=1k, 2.5k, 25k).

### J.4 GFP

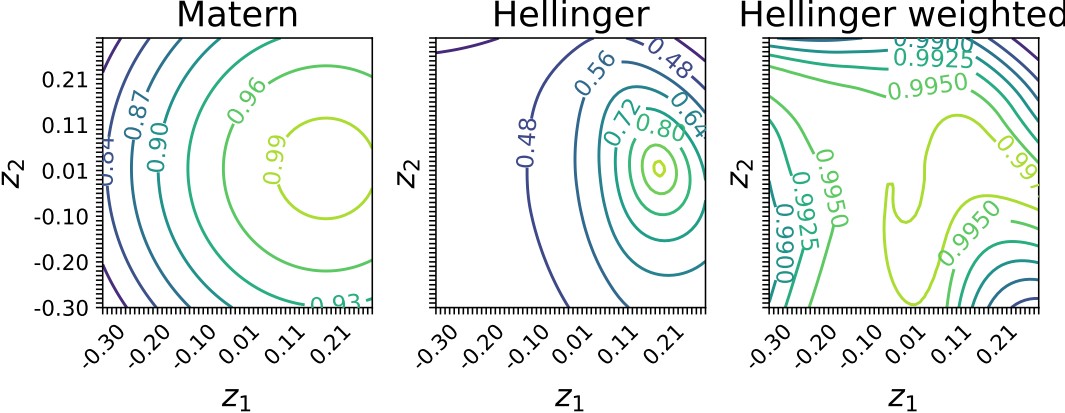

Figure A2: Covariance function values for the GFP (2D) latent space where the reference point is the GFP wild-type sequence. Comparing Matérn 5/2 with the (weighted) Hellinger kernel. The reference points corresponds to the *start* point in Fig. A1.

## J.5 RFP

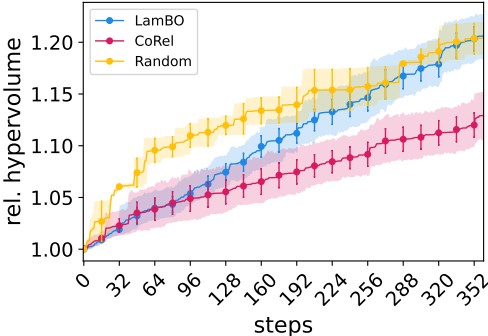

Figure A3: Discretely optimizating RFP we compare against LamBO in the warm-start setting. Starting N=50, batch-size=16 across seven seeds (random two seeds). Markers indicate batch averages with std.err. bars. Shaded region is 95% CI.

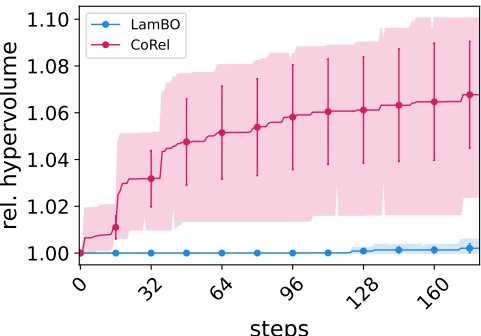

Figure A4: Optimizing RFP discretely in the reference case with 512 starting sequences. We compare `CoRel` against LamBO in the reference setup, batch-size=16 across three seeds. Given that we start with a relatively large starting hypervolume only marginal improvements can be achieved. Markers indicate batch averages with std.err. bars. Shaded region is 95% CI.

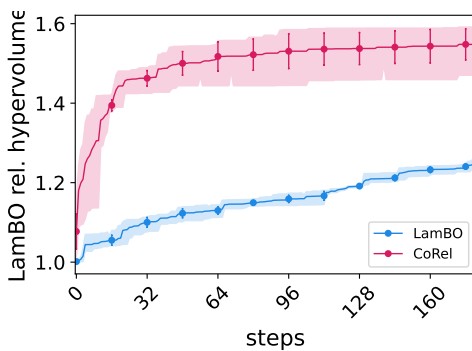

Figure A5: Optimizing RFP discretely in the reference case with 512 starting sequences using the LamBO specific relative hypervolume improvement. This metric is computed with internal reference Pareto front values, which remain fixed across all experiments and display a larger relative improvement over time. Batch-size is 16 across three seeds. Markers indicate batch averages with std.err. bars. Shaded region is 95% CI.

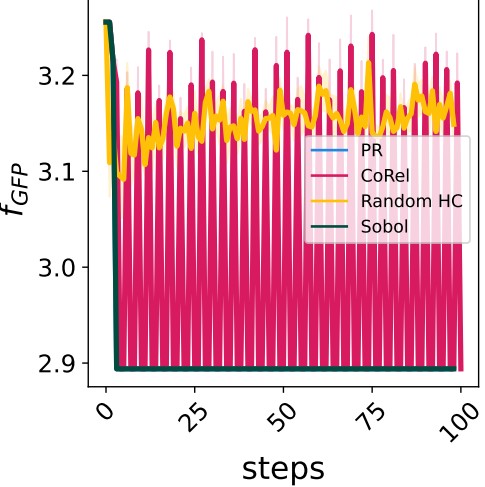

Figure A6: Oracle observations during the course of 100 GFP optimization steps. `CoRel` jumps between extreme values in contrast to the random climbing. Proposals by Sobol or PR sampling yield consistently subpar values.

