# OpenReview forum: "Bayesian Optimization over Discrete Structured Inputs by Continuous Objective Relaxation"
_TMLR — Rejected by TMLR_

### Review · Reviewer_MbhF · 2025-07-28

**Summary Of Contributions:**

The paper proposes a continuous relaxation method for Bayesian optimization (BO) in a discrete space. The basic idea is to construct a surrogate Gaussian process (GP) in a space of probability distribution over the (discrete) input domain. By incorporating a (pre-computed) generative distribution into the kernel function, the surrogate model can exploit similarity measure implied by the input distribution. The authors provide an algorithm that optimizes an acquisition function  defined over distribution space.

**Audience:**

Yes

**Claims And Evidence:**

No

**Requested Changes:**

In the discussion after (7), I have several questions. These are seemingly essential for the justification of the proposed method, but I do not think that currently sufficient explanations are provided. Could you elaborate at least the following questions?
- Why is 'sequences with high weighting are more independent' reasonable? To be honest, I currently do not think so.
- Why is the weighted distance is particularly suitable to the case with a threshold?
- What does 'reasonable correlation between the weighting and objective' mean? Why it can be assumed?

In Sec3.5, the authors mentioned that the well-known GP-UCB regret analysis is applicable. Then, what is the upper bound of information gain for Hellinger kernel？

What is acquisition functions in other compared methods? How are they optimized?

VanillaBO seemingly sufficiently good in Table 1 compared with CoRel (VanillaBO is better for discover, mpo, and other). How is this explained? It seems that Table 1 does not show the sufficient effectiveness of CoRel in terms of comparison with Vanilla. How is the advantage of the proposed method is demonstrated in this result.

Overall, many unclear descriptions exist, by which I felt difficulty to evaluate the paper. For example, please clarify
- In the second last paragraph of Sec2.1, the authors mentioned ``neither constrained probability space nor prior likelihood''. What do these mean? What do constrained probability space and prior likelihood mean?
- Is E_GP in Sec3.2 mean the expectation wrt posterior? (predictive distribution conditioned on training data) The definition is missing.
- After (5), r(1_x,1_x') = 1 is only when x \neq x'? It should have been noted.
- After (7), this should be 1/2 (w(x) + w(x'))? Otherwise, it becomes asymmetric.
- In Sec3.2.1 'Kernel specifications': What does 'Z_n as subsets of the latent space' mean? What is k_P(X|Z_n)? Why it is defined as a product of N points of the pre-training dataset? I read Z_n is for example VAEs latent training samples in Sec4, but it does not resolve these questions.
- In ii) of Sec3.3: What does 'Dec: ... exp^z_l,a'' mean?
- In Algorithm 1, what is LVM \phi? It suddenly appears without giving the detail.

Minor: Please add a reference for EHVI.

**Strengths And Weaknesses:**

S: The problem setting is important. Discrete (and sequence) data is still difficult to deal with by existing BO.

W: Technical descriptions are often quite vague, by which I still do not fully understand the implication of some claims in the paper.

W: Although the continuous problem would be easier than the original discrete problem, the dimension of the relaxed problem is high (L \times A, which typically can be more than several thousands) and so still can be quite difficult. Further, accuracy of the relaxed problem (how good the solution obtained through the relaxed problem in terms of the original discrete problem) is unclear.

---

> ### Author Response · Authors · 2025-09-11
>
> We thank the reviewer for their thorough review of our manuscript. We provide a point-by-point response below.
>
> #### Problem space size
>
> > Although the continuous problem would be easier than the original discrete problem, the dimension of the relaxed problem is high (L \times A, which typically can be more than several thousands) and so still can be quite difficult. Further, accuracy of the relaxed problem (how good the solution obtained through the relaxed problem in terms of the original discrete problem) is unclear.
>
> The total relaxed problem space is indeed large for moderately sized inputs and few tokens, even under factorization, which is why we require the weighting by a prior (domain) model to prioritize inputs. We agree with the reviewer that the size of the problem-space is a relevant concern, as are estimates of the empirical results in different regions of the problem space.
> However, the solutions of the proposed approach not only depend on the problem, but also on the relaxation and weighting. It is therefore an open issue to find an informative prior for the problem to be solved. This makes a comparison of the discrete problem versus relaxed problem dependent on the choices for both.
> We have clarified this in the Discussion paragraph on prior models:
>
> “The assumption that a well-defined model exists for the problem domain is crucial and our ability to find good solutions depends on the choices for relaxation parameterization and weighting function.
> That means if no prior model over the input tokens exists with which to derive a probability distribution, the proposed problem transformation is not solvable.“
>
> ### Requested Changes
> #### Independence of dissimilar sequences
>
> > Why is 'sequences with high weighting are more independent' reasonable? To be honest, I currently do not think so
>
> We thank the reviewer for pointing out that our earlier phrasing was unclear. Under the Hellinger kernel, dissimilar sequences correspond to orthogonal probability vectors, which yield a distance of 1 and thus zero similarity. In this sense, they contribute independently to the kernel matrix. If such sequences also carry high weights, their influence is affected by correlations with other points, and they act independently. In contrast, similar sequences (with nonzero overlap) share correlation structure, so that observing one reduces the effective weight of its neighbors. Importantly, *in the low-data regime we target, such strong modeling assumptions are necessary*: without them, the optimization problem is underdetermined, and we cannot avoid redundant evaluations or efficiently identify promising candidates.
>
> > Why is the weighted distance is particularly suitable to the case with a threshold?
>
> Our optimization setting is often threshold-driven: candidates are evaluated relative to a functional cutoff (e.g. active vs. inactive). In this case, once a sequence within a correlated region of sequence space is observed, the kernel structure ensures that the whole region is effectively discounted—either as promising (if above threshold) or unpromising (if below). Thus the weighted Hellinger kernel naturally aligns with threshold-based objectives by preventing redundant sampling of similar sequences and allowing the weighting to prioritize genuinely distinct, high-likelihood candidates.
> To clarify this point and the previous we have added the following under Proposition 3:
>
> “Because our setting is data-limited, we require strong modeling assumptions to make optimization feasible; the independence assumption for dissimilar, high-weight sequences is one such choice.”
>
> > What does 'reasonable correlation between the weighting and objective' mean? Why it can be assumed?
>
> We rely on the domain model for ranking and if the model likelihoods do not correlate with the objective they are not useful for weighting. This is established practice for modeling biochemical tasks and for example benchmarks like ProteinGym (Notin et al) test this and provide an overview of models that are suitable for ranking protein proposals. Specifically, a VAE where the ELBO correlates to a particular property can be used to optimize that property (as done in Goméz-Bombarelli et al 2018). On the flipside, using domain model likelihoods or weightings, which are uncorrelated to the objective yields a misspecified model because the rankings are not in agreement with the objective. The correlation of weighting with the target objective function is therefore a requirement for the approach to be feasible.
> We have clarified this in the referenced paragraph by adding :
>
> “In practice, we rely on the assumption that domain-model likelihoods correlate with objective values, an assumption supported by established benchmarks such as ProteinGym (Notin et al 2023), where likelihoods have been shown to provide meaningful rankings of candidate sequences.”

---

> > ### Author Response · Authors · 2025-09-11
> >
> > > the authors mentioned that the well-known GP-UCB regret analysis is applicable. Then, what is the upper bound of information gain for Hellinger kernel？
> >
> > We thank the reviewer for this inquiry and additional theoretical results. Since the contribution is a weighted Hellinger kernel the analysis needs to account for the weighting, which under the standard conditions, it does not. We have therefore clarified this by stating the following:
> >
> > “We note, however, that the regret analysis of Srinivas et al does not directly extend to our weighted Hellinger kernel, which is data-dependent and would require a new analysis of its RKHS and information-gain properties.
> > Developing such bounds is beyond the scope of this work, and in practice our algorithm employs Expected Improvement rather than UCB, for which only limited asymptotic convergence results exist; we therefore focus on empirical evaluation.”
> >
> > > What is acquisition functions in other compared methods? How are they optimized?
> >
> > We thank the reviewer for requesting the specifications for the baselines which use acquisition functions. The PMO comparisons follow the implementations in González-Duque et al. : PR and Bounce use EI, Turbo uses Thompson sampling, VanillaBO (following Hvarfner et al) uses log-noisy-expected improvement. For RFP LamBO uses NEHVI. For GFP PR uses EI.
> > We have added these specifications under the respective methods in appendix I.
> >
> > > VanillaBO seemingly sufficiently good in Table 1 compared with CoRel (VanillaBO is better for discover, mpo, and other). How is this explained? It seems that Table 1 does not show the sufficient effectiveness of CoRel in terms of comparison with Vanilla. How is the advantage of the proposed method is demonstrated in this result.
> >
> > We thank the reviewer for carefully assessing model comparisons in Table 1. While our model performs well for the protein tasks, it is competitive for two out of 25 small molecule (PMO) tasks (Table A3); and one task-aggregate (Table 1). The best BO method is Turbo which also performs best in one task aggregate (Table 1), while VanillaBO performs well in ‘other’ and ‘mpo’ tasks (Table A3). This empirical performance, for CoRel, can be explained by the dependence on the prior model and the latent optimization; suggesting that we require a better prior (domain) model, particularly for weighting.
> > We do acknowledge that we do not have SotA results on all empirical test cases. More work is required to find good prior models for small molecules.
> > We now clarify this in the discussion of the results:
> >
> > “However, CoRel is not consistently competitive across all tasks, suggesting that different, potentially task-dependent, priors should be investigated for small molecules.”
> >
> > > Sec2.1, the authors mentioned ``neither constrained probability space nor prior likelihood''. What do these mean? What do constrained probability space and prior likelihood mean?
> >
> > We have reformulated the particular sentence, which only referenced later propositions, contrasting our contributions to PR:
> >
> > “PR differs from our approach as it does not consider a constraint probability space of the inputs (which we introduce in Eq. (1) and (2)) and does not include likelihoods of the inputs given a parameterized prior model; ”
> >
> > > Is E_GP in Sec3.2 mean the expectation wrt posterior? (predictive distribution conditioned on training data) The definition is missing.
> >
> > We thank the reviewer for requesting a clear definition.
> > The complete clarified statement, is now :
> >
> > “Generally, it is not the case that the posterior mean of a GP ($E_{\text{GP}}[\bar{f}(\vec p)]:=E[\bar{f}(\vec p)|\mathcal{D}]$ for some dataset $\mathcal{D}$) is equal to the weighted sum of posterior means over each atomistic (Dirac) distribution $E_{\text{GP}}[\bar{f}(\vec p)]\neq\sum_{x\in\mathbb{X}}\vec p( \vec x )E_{\text{GP}}[\bar{f}(1_x)]$”.
> >
> > We emphasize that the standalone statement only asserts that in general this posterior mean is not equal to the weighted sum of posterior means over Dirac masses.
> >
> > > After (5), r(1_x,1_x') = 1 is only when x \neq x'? It should have been noted.
> >
> > We thank the reviewer for this correct observation, in accordance with the preceding “distinct input pair”. We have now added “$x\neq x’$” to clarify this distinction.
> >
> > > After (7), this should be 1/2 (w(x) + w(x'))? Otherwise, it becomes asymmetric.
> >
> > We thank the reviewer for this correction. We have fixed this.

---

> > > ### Author Response · Authors · 2025-09-11
> > >
> > > > In Sec3.2.1 'Kernel specifications': What does 'Z_n as subsets of the latent space' mean? What is k_P(X|Z_n)? Why it is defined as a product of N points of the pre-training dataset? I read Z_n is for example VAEs latent training samples in Sec4, but it does not resolve these questions.
> > >
> > > The kernel under Sec.3.2.1 par. “kernel specifications” is a product kernel over the pre-training data samples, and therefore defined as a product. The subscript refers to the weighting by the individual sample where p(X|Z_n) is the latent decoder conditioned on the latent encoding of the individual sample latent vector Z_n=enc(X_n). Each training sample we can encode via the given LVM, such that the set of latent vectors ({Z_n}) are subsets of the latent space.
> > > We have added a footnote clarifying that Z_n are latent-encoded inputs:
> > >
> > > “Specifically, given some latent encoder $\operatorname{enc}:\mathbb{X}\mapsto \mathbb{R}^D$ we obtain latent vectors $Z_n=\operatorname{enc}(\vec x)$ with which to construct $P(X|Z_n)$.”
> > >
> > > > In ii) of Sec3.3: What does 'Dec: ... exp^z_l,a'' mean?
> > >
> > > We thank the reviewer for requesting clarification of this condensed notation. We have now reformulated Sec. 3.3 ii) as :
> > >
> > > “a continuous parameterization of $\mathbb{P}$, a) given a ($D$-dimensional) probabilistic decoder mapping latent vectors $\vec z\in\mathbb{Z}^D$ to the space of factorizing distributions \,  $\operatorname{Dec}: \mathbb{R}^D \mapsto \mathbb{P}$,
> > >     and b) identifying each $P\in\mathbb{P}$ through the canonical softmax mapping $\operatorname{Dec}: \mathbb{R}^{L\times A} \mapsto \mathbb{P}$ with $\vec z_{l,a} \mapsto \frac{\exp(\vec z_{l,a})}{\sum_{a'=1}^A \exp(\vec z_{l,a'})}$ for $l\in[1,\dots, L]$ and $a\in[1,\dots, A]$, which enables a continuous optimization of $\beta(\vec z) \ce \alpha(\operatorname{Dec}[\vec z])$“
> > >
> > > > In Algorithm 1, what is LVM \phi? It suddenly appears without giving the detail.
> > >
> > > We thank the reviewer for requesting a re-introduction of $\phi$, which is in Sec. 2 “Problem statement”. We have added a clarifying remark prior to Alg. 1 :
> > >
> > > “The acquisition function is queried to find the maximizing probability vector with the predictive GP on the parameterized space of $D$ dimensions, given some probabilistic parameterization $\phi$.”
> > >
> > > > Please add a reference for EHVI.
> > >
> > > Thanks. We have added Daulton et al (2020) as a reference for the EHVI acquisition function.

---

### Review · Reviewer_NucT · 2025-08-04

**Summary Of Contributions:**

This paper proposes a new method for Bayesian optimization that supports discrete inputs via mapping them onto the space of probability distributions and performing optimization in this latent space. The proposed method is motivated by the setting with a lack of a large pool of labeled training examples and a limited black-box evaluation budget. The paper uses protein design as a central motivation for the problem where a sequence of discrete tokens (= amino acids) is mapped to a continuous value (e.g. thermal stability) that needs to be optimized.

The proposed method relaxes the original optimization problem to a continuous formulation by optimizing the objective under the probability distribution over all sequences up to a given length. The proposed probability distribution is further restricted to $P_f$ to enable computational efficiency & a specialized kernel operating on the probability distribution is defined which uses prior knowledge to rank sequences. The method also requires some separately trained mapping from $R^D \rightarrow P_f$ such as a VAE or HMM.

The authors apply their method to three protein or small molecule optimization tasks with public datasets and demonstrate superior performance on some tasks.

**Audience:**

Yes

**Claims And Evidence:**

No

**Requested Changes:**

## Critical questions
- Does the set $X$ enumerate all sequences or all tokens that make up sequences? After defining $\boldsymbol{p}$ in Eq. 1, does the example of $\boldsymbol{p}_{x} = 1$ imply that $x$ is a sequence or a single token? In other words, what does each component of $\boldsymbol{p}$ represent? Adding these examples to the text would make the construction of this distribution clearer.

- > the majority of the initial P space entries are highly unlikely given that inputs are of a particular length and tokens can be positionally conserved

Could you please clarify what "positionally conserved tokens" means and how that relates to the low probability regions of $P$?

- In Eq. 2, $P_f$ is defined using $\boldsymbol{p}_{l,a}$. Previously $\boldsymbol{p}$ was defined as a vector with a single subscript; how does one interpret two subscripts $l, a$? It appears that the constrained distribution considers sequences of length $l$ where all elements apart from the last one are 0 in the corresponding probability vector, and the last element is a sum of probabilities of each possible token occurring there. Is that the correct interpretation? Please clarify how $P$ and $P_f$ are to be interpreted.

- Section 3.2.1: Why is Hellinger kernel used? How does it "exploit the structural properties of $P_f$" and why do other kernels don't work for the problem at hand? A slower introduction of the Hellinger kernel that uses visualization in Fig.1 would be helpful for the reader. Also, please show why Hellinger kernel is 1 for any distinct input pair; are samples from $P_f$ also represented as $1_x$?

- >  there often exists a prior ranking over elements of X in form of a probability distribution

If there is already a probability distribution over elements of $X$ that we will utilize for computing weights for the kernel, why do we need to construct distributions $P$ and $P_f$?

- After Proposition 3: Is the expression for $r_w$ missing brackets, i.e. should $1/2$ be a multiplier for the sum? Also is this the expression for $r$ or $r^2$?

>  Brookes et al. (2019) provide a continuous parameterization of Pf to solve the GFP problem in the form of a pre-trained latent decoder
- Is the representation $P_f$ novel to this work or has it been introduced before? Please cite the corresponding work under the "Representation" section if the latter is the case.

- Section 4.1 introduces "CoRel" in passing whereas it appears to encompass the proposed method. Introducing it slower and explicitly stating its novelty and steps would help make this section clearer.

- > Fig. 1 shows the evaluated Hellinger (Eq. (3)) and weighted Hellinger kernel (Eq. (6)), which are not equidistant in latent space, like the Matérn kernel evaluations

How does that demonstrate superior performance of the proposed method on this task?

- Why was CoRel not compared against VanillaBO and/or more methods in the GFP optimization task?

- >CoRel obtains the best performance for logP optimization and the gsk3 docking task compared to other methods.

Turbo appears to be the best in the optimize task

- Please explain how CMA-ES is used in the references regime for the PMO task. Also, this method achieves the best performance on most tasks: please comment on that.


## Other questions
- The authors mention that the novelty of their work lies in using a continuous kernel for distributions over discrete sequences and defining a GP prior over the objective. Could you comment on how the works on multi-objective BO using scalarizations (Paria et al. 2019, Selega & Campbell, 2024) where distributions over scalarization weights can be seen as setting priors over objectives, relate to the stated novelty?

- Section 3.1: Optimization. Was $\alpha_m$ defined yet? Which acquisition function will be used?

- Section 3.2: Please define "an atom" in the context of GPs.

- Eq. 6: font of $p,q$ switches compared to previous equations; is that intended?

- > We set λ and σ2 by maximizing the evidence log p(y|λ, σ2)

Are the hyperparameters set over a validation set?

- Algorithm 1: Define $D, b$

- Algorithm 2: line defining $y^*$ should say $\alpha$ not $a$

- > A discrete optimizer together with UCB as the acquisition function can achieve sublinear regret

Section 3.5 mentions UCB but all experiments use EI as the acquisition function. Please comment on this discrepancy.

- > However, we find that if significantly more starting candidates are available, a larger Pareto front is optimized and LamBO outperforms our method.

Please add or cite an appendix figure demonstrating that

**Strengths And Weaknesses:**

### Strengths
- The work proposes a novel computationally tractable BO method that can operate in a challenging setting of discrete element sequences where the space of possible solutions is combinatorially exploding
- The method is motivated by an important real-world application of molecular design and is specifically tailored to the commonly occurring scenario of limited training data and evaluation budget

### Weaknesses
- Though the paper is written well at the beginning, it quickly becomes opaque as it builds the proposed framework, including in the core definitions; significant clarifications and/or examples are needed to improve readability
- Empirical results are introduced hastily such that readers not already familiar with the used tasks will have a hard time following the experiments
- The introduction states that the proposed method performs "remarkably well" yet the results (or the discussion of the results) do not make this claim obvious
(e.g. CoRel performs best only on 1 task in Table 1; for GFP, it is stated "CoRel prioritizes extreme values of the oracle" -- is that desirable?; for RFP, another method performs better with more input data)

---

> ### Author Response · Authors · 2025-09-11
>
> We thank the reviewer for their thoughtful and thorough review. We appreciate that the novelty and tractability of our approach are highlighted, together with “important real-world application” for limited data scenarios. We address the weaknesses and critical inquiries in turn below.
>
> ### Weaknesses
> We thank the reviewer for the requested clarifications which we have now added, as described in the points below.
>
> > Empirical results are introduced hastily such that readers not already familiar with the used tasks will have a hard time following the experiments.
>
> We have reformulated the introduction to Sec. 4 to provide more context for the referenced tasks. This now includes:
>
> “The first task (RFP) requires us to optimize both the stability and surface accessibility of protein candidates by modifying the tokenized sequence of amino acids and evaluates established domain oracles (see appendix I.3).
> The second (GFP) task evaluates the fitness surrogates in Brookes et al.
> (2019) as a proxy for green protein fluorescence.
> The last set of multiple tasks (PMO) are a benchmark for label-efficient small molecule optimization based on property optimization, (re-)discovery of molecules, docking proxies, and other tasks.“
>
> This information has so far only been available in appendix Sections H and I .
>
> > The introduction states that the proposed method performs "remarkably well" yet the results (or the discussion of the results) do not make this claim obvious (e.g. CoRel performs best only on 1 task in Table 1; for GFP, it is stated "CoRel prioritizes extreme values of the oracle" -- is that desirable?;
>
> We thank the reviewer for pointing out that the formulation “remarkably well” is *incorrect in the context of all small molecule optimization* results. We have now corrected this to:
>
> “we demonstrate that our proposed approach performs well when optimizing protein sequences and is on par with established BO methods in optimizing small molecule properties under a very tight evaluation budget.”
>
> For the GFP values we merely state the observed empirical behavior, since CBas is a (deep-learning) derived proxy, as opposed to the FoldX computed stability in RFP. Appendix I.4 gives an overview over the values in a dimensionality reduced space. Under a potentially erratic oracle, repetitively querying for extreme (high) values is desirable, as opposed to the observed non-performance of the baseline methods.
>
> > for RFP, another method performs better with more input data
>
> The challenge for the empirical evaluation is that only few starting observations are given. It is expected that other methods, e.g. LamBO, outperform our method, with significantly more initial training data.
>
> #### Definitions
>
> > Does the set X enumerate all sequences or all tokens that make up sequences? After defining p in Eq. 1, does the example of p_x=1 imply that x  is a sequence or a single token? In other words, what does each component of p represent? Adding these examples to the text would make the construction of this distribution clearer.
>
> The set X is the space of all sequences, which are composed of all tokens that make up a sequence. Therefore, $\vec p_x=1$ is the likelihood of the complete sequence. In this context, the indicator is the evaluation of all the tokens, which comprise a sequence, having their expected values. Each component of $\vec p$ is therefore the likelihood of a given token. We thank the reviewer for pointing this out and have rephrased the key sentences in Sec. 3.1 to:
>
> “Note that each element of $\vec x \in X$, which is a sequence of tokens, can be represented as ${\vec p := 1_{\vec x}}$; i.e. as Dirac probability vector over the tokens with full mass on $\vec x$, such that ${\vec p}_{
> \vec x'
> }=1$ iff $\vec x=\vec x'$ and $0$ otherwise.“

---

> > ### Author Response · Authors · 2025-09-11
> >
> > > Could you please clarify what "positionally conserved tokens" means and how that relates to the low probability regions of P?
> >
> > In the context of our practical applications, we expect inputs to adhere to specific properties. That is, they have a particular length: (our) proteins have 200-400 positions and small molecules are tokenized to 25-100 elements. Therefore finding inputs that are an order of magnitude shorter or longer is highly unlikely given the problem setting and the prior model over inputs. Additionally, for inputs of a particular problem we expect a particular set of tokens in specific places (e.g. we can define a GFP or RFP by recurring tokens in specific positions).
> > To clarify this we have now added
> >
> > “For example, a protein to be optimized has a particular length and is characterized by the occurrence of specific, positionally conserved tokens.” preceding Eq. 2.
> >
> > While this is true for optimizing proteins, this is not strictly true for all discrete problems of particular length. We have now added this limitation to the discussion as:
> >
> > “Furthermore, not all discrete problems can assume inputs of consistent length and positional, problem-specific conservation of tokens -- that is, consistent structure over inputs. Both the length consistency and a prior model over token occurrence are requirements for our approach.“
> >
> > > In Eq. 2, P_f  is defined using p_l,a. Previously p was defined as a vector with a single subscript; how does one interpret two subscripts l,a ? It appears that the constrained distribution considers sequences of length l  where all elements apart from the last one are 0 in the corresponding probability vector, and the last element is a sum of probabilities of each possible token occurring there. Is that the correct interpretation? Please clarify how P and P_f  are to be interpreted.
> >
> > We go from a space of probability vectors over all possible inputs to the space of factorizing distributions – ie. Eq.1 to Eq.2. That means that in $P_f$ (Eq.2) each position sums to 1, whereas previously the sum over all inputs is one (Eq.1). We therefore require the second subscript over the length of the sequences $l \in L$, such that we sum over all categories at a particular position $a \in | A|$.
> > We thank the reviewer for inquiring about the notational differences here - we have now clarified under Eq. 2 that:
> >
> > “This effectively gives us a matrix of likelihoods that at each position sum to one and which we index by position over length and tokens, respectively. “
> >
> > > Section 3.2.1: Why is Hellinger kernel used? How does it "exploit the structural properties of P_f" and why do other kernels don't work for the problem at hand? A slower introduction of the Hellinger kernel that uses visualization in Fig.1 would be helpful for the reader. Also, please show why Hellinger kernel is 1 for any distinct input pair; are samples from P_f also represented as I_x?
> >
> > We require a kernel that is (i) positive semi-definite, (ii) efficient to compute, scaling to large numbers of samples, and (iii) can be weighted with likelihoods., The Hellinger kernel satisfies these criteria. It is directly defined for the space of factorizing distributions $P_f$ and therefore directly leverages the probability vectors as described in Sec. 3.2 . While other metrics over probability vectors exist (e.g. Jensen Shannon Divergence, Wasserstein-1 distance), their induced kernels are usually not as efficient to compute or to weight - making them impractical for our setting.
> > We have reformulated and extended the discussion par. “Other distance measures apply” (Sec. 5) to include why the Hellinger kernel is used :
> >
> > “To be able to utilize a kernel we require it to be (i) positive semi-definite, (ii) efficient to compute, such that it scales to many samples, and (iii) it can be weighted with likelihoods. The Hellinger kernel satisfies these criteria. It is directly defined for the space of factorizing distributions $P_f$ and therefore directly leverages the probability vectors as described in Sec. 3.2 . While other metrics for probability vectors exist (e.g. the Jensen Shannon Divergence or Wasserstein-1 distance), their induced kernels are usually not as efficient to compute or to weight - making them impractical for our setting.
> > CoRel can potentially work with either; however we emphasize the linear runtime of our kernel, which may not translate to alternatives.”
> >
> > The distance evaluation to 1 (pg. 5, Eq.5) for distinct input pairs, follows directly from orthogonal inputs for which we obtain $r(x,x’)=1-0$. For example, $r(AB; BA) = 1 - \prod_1^2 \sum_{A,B}\sqrt{[1,0;0,1] [0,1;1,0]} = 1 - 0 = 1$ . We’re happy to include this short example in the appendix.
> > Finally, samples from P_f are realizations of sequences. Any sequence we can denote via an indicator. Therefore, yes, samples from P_f can be indicator functions.

---

> > > ### Author Response · Authors · 2025-09-11
> > >
> > > > If there is already a probability distribution over elements of X that we will utilize for computing weights for the kernel, why do we need to construct distributions P and P_f ?
> > >
> > > A positive weighting of inputs does not necessarily imply that it can be used to parameterize the distribution - i.e. it is necessary but not sufficient. In general weights can be derived from many sources and need not correspond to a normalized distribution with the properties we require. The role of $P_f$ is precisely to define such a parameterized distribution over X. It is true that in the examples we provide, the weights are derived from the distributions, but in principle the two objects can be decoupled.
> > > We have clarified this in the revised text under par. “Valid distributions w”:
> > >
> > > “Importantly, the weighting distribution can be distinct from the distribution objects in $\mathbb{P}_f$.
> > > While in this work the weighting and the model that parameterizes the representation are the same, this is not a requirement.“
> > >
> > > > After Proposition 3: Is the expression for r_w missing brackets, i.e. should 1/2 be a multiplier for the sum? Also is this the expression for r or r^2 ?
> > >
> > > The expression (Prop.3) is as stated - with ½ applied to the sum. For the derivation with the Hellinger distance $r^2$, see Appendix D.1 for details.
> > >
> > > > Is the representation P_f novel to this work or has it been introduced before? Please cite the corresponding work under the "Representation" section if the latter is the case.
> > >
> > > The construction of *$P_f$ via parameterizing distributions is a novel contribution of this work*. Specifically, we introduce $P_f$ as a structured representation that enables kernel-based methods to operate directly on factorizing distributions. The specific case of applying our framework to the GFP problem involves the use of a latent variable model (Brookes et al) as an example pre-trained probabilistic decoder that we can use to practically construct $P_f$.
> > >
> > > We note, however, that at its core $P_f$ is a special form of integer relaxation and thus conceptually relates back to ideas in the optimization literature of the 1980s-1990s, which we cite.
> > >
> > > > Section 4.1 introduces "CoRel" in passing whereas it appears to encompass the proposed method. Introducing it slower and explicitly stating its novelty and steps would help make this section clearer.
> > >
> > > Section 3.4 introduces “CoRel” with the key algorithms following the introductions in Sec. 3, whereas Sec. 4 relies on specific model choices to obtain continuous and discrete optimization results. We thank the reviewer for pointing out that novelty contributions can be clarified and we have added a summary of to the end of Sec 3.4:
> > >
> > > “The key contributions in the two algorithms are (i) the use of factorizing distributions through a prior model $\phi$, (ii) the surrogate model with a prior weighting, and (iii) the acquisition function acting on $\mathbb{P}_f$.
> > > While the use of an $\arg\max$ can be considered a common choice to recover a sequence, our approach allows us to sample sequences from the distribution object.”
> > >
> > > #### Empirical evaluations
> > >
> > > > How does that demonstrate superior performance of the proposed method on this task?
> > >
> > > We do not claim superior performance in the qualitative investigation and Fig. 1 visualizes the difference of the Hellinger distance before and after weighting; effectively not assigning equal (equidistant) importance compared to the Euclidean distances like the Matern kernel, which is the intended effect.
> > >
> > > > Why was CoRel not compared against VanillaBO and/or more methods in the GFP optimization task?
> > >
> > > The GFP task tests our method against three baselines: a contemporary method PR, Sobol sampling, and random HillClimbing. Due to the implementation of the objective function and latent-optimization in a TensorFlow1 framework a test against torch based methods e.g. Bounce, VanillaBO, etc. require substantial additional implementations. Therefore the comparison against more BO methods has been done in the PMO comparison on the 25 tasks on small molecules with a consistent (PyTorch) backend.
> > >
> > > > Turbo appears to be the best in the optimize task
> > >
> > > We thank the reviewer for this correction. CoRel performs best for “qed”, not “logP”, and Turbo indeed performs best for “logP. We have corrected this now.

---

> > > > ### Author Response · Authors · 2025-09-11
> > > >
> > > > > Please explain how CMA-ES is used in the references regime for the PMO task. Also, this method achieves the best performance on most tasks: please comment on that.
> > > >
> > > > The CMA-ES algorithm acts on the latent space directly, leveraging its covariance structure, and its performance is an interesting result. It appears that while the dimensionality of the latent (continuous) optimization is high enough to be challenging for BO methods, e.g. PR, Bounce or VanillaBO, the latent model correlates well with the target objective, which is directly leveraged in the CMA-ES algorithm.  The optimization budgets are exactly the same as for the other algorithms. The algorithm settings are exactly the ones as presented in (Gonzalez-Duque et al) and we have added a clarifying section to the appendix (Sec. I.11). We have added a paragraph with discussion of the results (I.11) in the revised version.
> > > >
> > > > #### Other questions
> > > > > The authors mention that the novelty of their work lies in using a continuous kernel for distributions over discrete sequences and defining a GP prior over the objective. Could you comment on how the works on multi-objective BO using scalarizations (Paria et al. 2019, Selega & Campbell, 2024) where distributions over scalarization weights can be seen as setting priors over objectives, relate to the stated novelty?
> > > >
> > > > We thank the reviewer for the additional related works on multi-objective BO. We have included the work by Paria et al and Selega and Campbell in the related works section. The additional examples do not contest our stated novelty, as these approaches do not consider a continuous relaxation of factorizing distributions. Even when extending their propositions to view their scalarization as domain priors over objectives, we still require constraints on the probability space and a weighting of the covariance function.
> > > > The related works section now includes :
> > > >
> > > > “Paria et al propose multi-objective Bayesian optimization by scalarizing functions of the objective and a Pareto front objective; in practice this scalarization is often linear.
> > > > This approach is extended in Selega and Campbell to expectations over the scalarized acquisition and applied to biomedical objectives.
> > > > Stanton et al also accounts for multiple objectives by optimizing a Pareto front.“
> > > >
> > > > > Section 3.1: Optimization. Was \alpha  defined yet? Which acquisition function will be used?
> > > >
> > > > The highlighted paragraph introduces acquisition $\alpha$ generally and any BO acquisition function applies. The specific choices are introduced in Sec. 4 given the optimization problems, for single objectives this is EI and for multi-objectives this is NEHVI.
> > > >
> > > > > Section 3.2: Please define "an atom" in the context of GPs.
> > > >
> > > > We thank the reviewer for requesting clarification on the use of *atom* in the context of GP posterior predictives. We have reformulated it to “atomistic (Dirac) distribution”, clarifying its definition.
> > > >
> > > > > Eq. 6: font of p, q  switches compared to previous equations; is that intended?
> > > >
> > > > We thank the reviewer for pointing out the font inconsistencies, which had not been intentional. This is now corrected.
> > > >
> > > > > Are the hyperparameters set over a validation set?
> > > >
> > > > The hyperparameter optimization is not done with a validation set, given that for the BO routine only few labels exist and no (external) validation set is available. The values are therefore computed on the available dataset at a given iteration.
> > > >
> > > > > Algorithm 1: Define D, b
> > > >
> > > > We thank the reviewer for the requested clarification. Apart from their earlier occurrence in the text, $b$ belongs to *budgets*, we have now defined D prior to Alg. 1 :
> > > >
> > > > “The acquisition function is queried to find the maximizing probability vector via the predictive GP on the parameterized space of $D$ dimensions.”
> > > >
> > > > > Algorithm 2: line defining y* should say \alpha not a
> > > >
> > > > Thanks. We have corrected this.
> > > >
> > > > > Section 3.5 mentions UCB but all experiments use EI as the acquisition function. Please comment on this discrepancy.
> > > >
> > > > It is correct that these theoretical results, to the best of our knowledge, only apply to acquisition with UCB and not necessarily EI and we use the latter for our empirical evaluation. In accordance with another review we have now reformulated this section :
> > > >
> > > > “A discrete optimizer together with UCB as the acquisition function can achieve sublinear regret under the conditions laid out in Srinivas et al (2012) .
> > > > We note, however, that the regret analysis of Srinivas et al does not directly extend to our weighted Hellinger kernel, which is data-dependent and would require a new analysis of its RKHS and information-gain properties.
> > > > Developing such bounds is beyond the scope of this work, and in practice our algorithm employs Expected Improvement rather than UCB, for which only limited asymptotic convergence results exist; we therefore focus on empirical evaluation.”
> > > >
> > > > > cite the appendix figure
> > > >
> > > > We thank the reviewer for this request. We have added a reference to appendix Fig. A3.

---

### Review · Reviewer_1WYB · 2025-08-28

**Summary Of Contributions:**

The paper addresses Bayesian optimization (BO) over large, discrete, structured spaces (e.g., protein or molecular sequences) under very small labeled budgets. The core difficulty is that gradients don’t exist for discrete inputs and most BO methods (surrogates and acquisition optimizers) are designed for continuous domains. The authors propose to relax the discrete objective into a continuous one by optimizing the expected objective over distributions, and then build GP surrogates directly over distributions using a (weighted) Hellinger kernel that can incorporate prior likelihoods from domain models. The authors empirically validate the model on a wide range of tasks.

**Audience:**

Yes

**Claims And Evidence:**

Yes

**Requested Changes:**

1. The restriction to factorizing distributions $\mathbb{P}_f$ is critical for computational tractability. This assumes positional independence in the sequences being optimized. How might this assumption limit performance on problems where strong, long-range interactions between positions are crucial for the objective function (e.g. complex ring structures in molecules)? It would be beneficial to discuss the potential limitations imposed by this choice.

2. Can you explain in more details of the intuition using product-of-kernels vs. other alternatives? Is the optimization stable? A product of many kernel values (which are typically $\le 1$) could lead to vanishingly small similarities for even moderately distant points. Could the authors provide more intuition for this choice over, for example, an averaged kernel? How does this formulation affect the GP's length scales, and how does its computational cost scale with the number of samples $N$?

3. The paper notes that the full product kernel on the GFP task was terminated after 60 hours for just 26 iterations. Can you provide some detailed analysis of computational complexity?

4. Algorithm 2 describes the process of selecting a discrete sequence from the optimized continuous distribution.. This involves taking the mode (argmax) and then potentially improving upon it by sampling. Could you clarify the role of the budget $b$ in this algorithm?

4. The generalizability of the prior model: How would the method perform if only a misspecified or weakly correlated prior model is available?

**Strengths And Weaknesses:**

1. Reformulating the problem as BO over distributions (not just over learned continuous embeddings) is elegant. The weighted Hellinger kernel is a natural way to inject calibrated prior knowledge while retaining linear‑in‑sequence‑length kernel evaluations.

2. The presentation of paper is clear and the related work is informative.

3. The experiments are realistic and informative.

---

> ### Author Response · Authors · 2025-09-11
>
> We thank the reviewer for their review and clarifying inquiries. We appreciate that the core contributions of the paper are highlighted as an “elegant” formulation of the problem with its use of prior knowledge, linear-in-sequence-length computations, and realistic experiments. We address the additional inquiries and requested changes below.
>
> #### Limitations from factorizing
> >The restriction to factorizing distributions  is critical for computational tractability. This assumes positional independence in the sequences being optimized. How might this assumption limit performance on problems where strong, long-range interactions between positions are crucial for the objective function (e.g. complex ring structures in molecules)? It would be beneficial to discuss the potential limitations imposed by this choice.
>
> The positional independence assumption, critical for computational tractability, works well for independent positions in protein sequence problems. The observation that this model requirement is limiting for long-range interacting positions is correct, i.e., a practical model assumption yielding an oversimplification.
> We have added two sentences to the discussion emphasizing this shortfall; specifically appending to the first paragraph of Sec. 5:
>
> “We obtain this computability, in part, due to the factorizing assumption underlying $\mathbb{P}_f$ (Eq.\cref{eq:factor}).
> While this modeling assumption holds for inputs with independent tokens, it fails for co-dependencies in the inputs, e.g. long-range residue dependence or small molecule substructures and is thus limited.”
>
> #### Product kernel implications
> > Can you explain in more details of the intuition using product-of-kernels vs. other alternatives? Is the optimization stable? A product of many kernel values (which are typically ) could lead to vanishingly small similarities for even moderately distant points. Could the authors provide more intuition for this choice over, for example, an averaged kernel? How does this formulation affect the GP's length scales, and how does its computational cost scale with the number of samples ?
>
> The choice of a product kernel translates the weighting of inputs directly to a ranking by the covariance function for similar sequences, e.g. for latent-variable models; such that if the ranking model likelihood yields a correct ranking of sequences the product kernel makes this available. The choice of using the product directly emphasizes the choice of the prior weighting.
> A product kernel with $N$ components requires $N$ evaluations of the base covariance per pair of points. We rely on the Cholesky factorization for inversion, which scales cubically with the size of the dataset. We have now clarified the this cost (Sec. 3.2.1 “kernel specifications”) as:
>
> “Given $N$ pre-training samples and $n=|\mathcal{D}_t|$ as the size of the dataset at a given timestep $t$ of the BO algorithm, we obtain $\mathcal{O}(n^2NL|A|)$ for the cost of the complete Gram matrix construction and the usual $\mathcal{O}(n^3)$ for the Cholesky decomposition when computing the posterior predictive.”
>
>
> Due to the impact of the pairwise construction we subsample the available components for computational feasibility.
> It is true that small likelihoods can, in practice, result in unstable optimization by numerical underflow. This potentially yields extreme hyperparameter values during optimization. Empirically, given a VAE with a few hundred dimensions and $N$ a few thousand components, this optimization is still computable. With more components and potentially smaller weights the stability can be recovered by computing the values in log-space instead. We have added a subsection noting this in appendix Sec. D “Stabilizing the product kernel weightings”.
> We thank the reviewer for their proposal to use an average kernel – assuming an arithmetic mean with (proportionally normalized) weights – which is an interesting additional model. This choice would place less emphasis or dependence on the weighting, making it less conservative, and should be less prone to vanishing weights.
> The choice between product over an average can intuitively be viewed as: if one trusts the prior weighting, take the product, else take a normalized average.
> To clarify this choice we have further appended to par. “kernel specifications”:
>
> “Lastly, the choice of a product kernel aggregate emphasizes the dependence on the weighting function.
> Other valid modeling choices exist that rely on different weighting contributions (Jebara et al 2004).”
>
> > Can you provide some detailed analysis of computational complexity?
>
> The computational complexity remains the same (see above). The long empirical run-times for full product kernel evaluations (e.g. $N=50 000$ instead of subsampled N) are due to computations not being fully optimized and CPU dependent. Converting all kernel evaluations to CUDA operations, including the weighting computations, should speed up runtimes significantly.

---

> > ### Author Response · Authors · 2025-09-11
> >
> > #### Sequence evaluation budgets
> >
> > > Algorithm 2 describes the process of selecting a discrete sequence from the optimized continuous distribution.. This involves taking the mode (argmax) and then potentially improving upon it by sampling. Could you clarify the role of the budget  in this algorithm?
> >
> > A larger internal budget in Alg. 2, results in more sequences being evaluated during sequence recovery. This can translate to higher performance when finding the discrete sequence. We have clarified this by reformulating (Sec. 3.4):
> >
> > “From the probability vector we can obtain a sequence or a set of sequences within a given budget (see Alg. 2). The larger the internal sampling budget ($b$) the more sequences proposals will be evaluated.“
> >
> > #### Prior dependence
> >
> > > The generalizability of the prior model: How would the method perform if only a misspecified or weakly correlated prior model is available?
> >
> > Given a misspecified prior or only weakly correlated weighting model results in no performance or low performance respectively. Under such conditions, in practice, we revert to an optimization akin to randomly sampling the objective. We thank the reviewer for clarifying this limitation and have added to the last discussion point (Sec. 5):
> >
> > “Furthermore, if the weighting function is only weakly correlated -- or entirely uncorrelated -- with the target function, then the kernel cannot guide candidate selection effectively.
> > In this case, the procedure reduces to essentially random sampling of the objective.”

---

### Decision · Action_Editor_uU3T · 2025-10-30

**Recommendation:** Reject

**Audience:**

Yes

**Audience Explanation:**

The current set of initial theoretical results are interesting, although the main theoretical result (sublinear regret bounds of the proposed algorithm) were found to not be applicable during the rebuttal.

**Claims And Evidence:**

No

**Claims Explanation:**

Reviewers agreed that empirical results were not sufficiently strong.  While the authors addressed reviewer criticisms (largely by improving writing clarity), a major theoretical contribution had to be walked back during the rebuttal when challenged by Reviewer MbhF.  As pointed out by other reviewers, the proposed method is not (nor does it have to be) SOTA, however, it should be somewhat competitive to vanilla BO.  For the former situation, ideally some practical advantage should be demonstrated which the continuous relaxation provides (e.g., interpretability that cannot be provided by vanilla BO).  In its current state, the paper has some interesting initial theoretical results, while requiring more practical advantages demonstrating that the proposed continuous relaxation may offer compared to related BO alternatives.

**Resubmission Of Major Revision:**

The authors may consider submitting a major revision at a later time.